# VIDEO REASONING WITHOUT TRAINING

## ABSTRACT

Video reasoning using Large Multimodal Models (LMMs) relies on costly reinforcement learning (RL) and verbose chain-of-thought, resulting in substantial computational overhead during both training and inference. Moreover, the mechanisms that control the thinking process in these reasoning models are very limited. In this paper, using entropy of the model's output as a signal, we discover that the high-quality models go through a series of *micro-explorations* and *micro-exploitations* which keep the reasoning process grounded (i.e., avoid excessive randomness while the model is exploring or thinking through an answer). We further observe that once this "thinking" process is over, more accurate models demonstrate a better convergence by reducing the entropy significantly via a final exploitation phase (i.e., a more certain convergence towards a solution trajectory). We then use these novel, theoretically-grounded insights to tune the model's behavior directly at inference, without using any RL or supervised fine-tuning. Specifically, during inference, our proposed approach called `V-Reason` (Video-Reason) adapts the value cache of the LMM via a few optimization steps on a small, trainable controller using an entropy-based objective, i.e., no supervision from any dataset or RL is necessary. This tuning improves the model's micro-exploration and exploitation behavior during inference. Our experiments show that our proposed method achieves significant improvements over the base instruction-tuned models across several video reasoning datasets, narrowing the gap with RL-trained models to within **0.6%** average accuracy without any training, while offering massive efficiency benefits: output tokens are reduced by **58.6%** compared to the RL model.

## 1 INTRODUCTION

Reasoning with generative AI models, such as Large Language or Large Multimodal Models (LLMs/LMMs), has gained substantial attention recently. This capability is implemented by asking the model to "think" about a problem, before making a final recommendation, and can be accomplished by several approaches, including Chain-of-Thought (CoT) (Wei et al., 2022), supervised fine-tuning with CoT (CoT-SFT) (Liu et al., 2025; Feng et al., 2025), or reinforcement learning (RL) with a *thinking-before-answering* format (Guo et al., 2025; OpenAI et al., 2024). Although initial progress was shown mainly for LLMs, such ideas have now been extended to video reasoning problems (Feng et al., 2025; Li et al., 2025; Zhang et al., 2025b; Cheng et al., 2025; Wang et al., 2024) by exploiting Vision-Language LMMs. Although successful, CoT-SFT, and RL-based methods tend to be highly computationally intensive, both for training and inference, due to the long thinking traces that they tend to produce. These costs are particularly exacerbated for video, due to the high resolution and multiple frames involved in the reasoning process. Furthermore, there remains little understanding of the factors that control the depth and quality of the reasoning process. In this paper, we seek to address these problems by considering the following **key questions**:

1. Can inference-time metrics characterize the thinking process of video reasoning models? If yes, can these metrics differentiate between higher- and lower-quality reasoning LMMs?
2. Can such metrics be used to formulate novel inference-time optimization objectives that enhance video reasoning without requiring additional model training?

To answer these questions, we first analyze the model's output distribution entropy at generation step $t$ computed as $H_t = -\sum_{i \in \mathcal{V}} p_t^i \log p_t^i$ ($\mathcal{V}$ refers to vocabulary of the model) for instruction-tuned LMMs of various sizes, as shown in Fig. 1(a). This analysis reveals two broad trends: (*i*) all models exhibit a pattern of increasing and then decreasing entropy as tokens are generated, and (*ii*) larger, more accurate, models have *lower and delayed entropy maxima*, followed by a reduction phase that converges to a *lower final entropy* (see Fig. 1(a) and its inset).

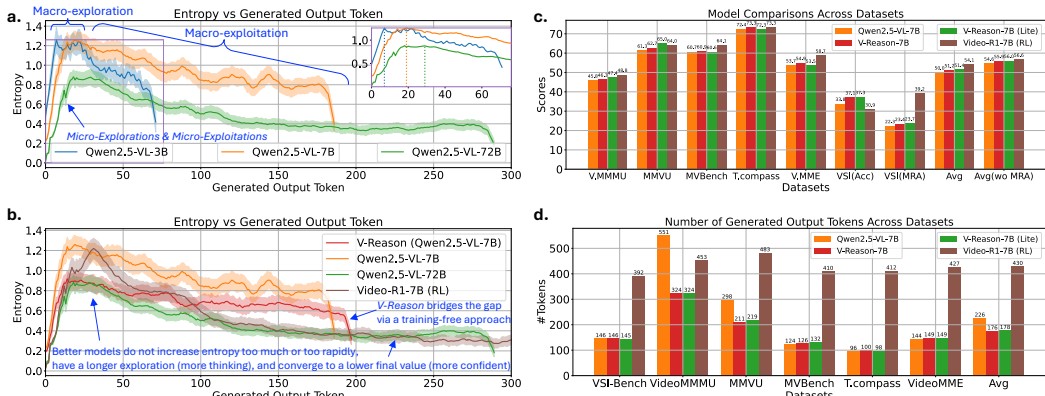

Figure 1: V-Reason Overview: (a) Entropy of the output distribution averaged over the MMVU (Zhao et al., 2025) dataset of 625 videos. We see clear macro-exploration and macro-exploitation phases with bigger, more accurate models showing lower overall entropy (lower and later peak, followed by a lower final entropy during the macro-exploitation). We use these key insights to adapt a model's behavior in a training-free way using an inference-time optimization technique. (b) Applying V-Reason on Qwen2.5-VL-7B-Instruct makes its entropy behave more similarly to the larger or the RL-trained Video-R1-7B model. (c) Our method achieves higher accuracy than the base LMM and bridges the accuracy gap with the RL model. (d) V-Reason also significantly reduces the total output tokens compared to all models due to a dedicated entropy minimization phase.

The first trend above can suggest a formal definition of the "thinking" in terms of output distribution entropy. As the model starts generating a response, it seems to be uncertain and *searches* through multiple solution trajectories, which can explain the increase in its output entropy. We denote this as the *macro-exploration* phase. As the generation progresses, the model seems to start to identify the correct thinking thread, and becomes increasingly certain about a solution, resulting in the gradual reduction in the entropy of its output. We denote this as the *macro-exploitation* phase.

The second trend seems to suggest that entropy should *not* increase too rapidly during the macro-exploration phase. In fact, all models go through a series of *micro-exploration* and *micro-exploitation* cycles (characterized by small increases and decreases of entropy) during both macro phases of the thinking process; see Fig. 1(a) shaded regions. A delayed entropy peak can suggest that better reasoning models explore more alternative answers, leading to longer thinking threads, which has been identified as a sign of better thinking in the literature (Wei et al., 2022; Guo et al., 2025; OpenAI et al., 2024). In this context, more and/or longer cycles of micro-exploration and micro-exploitation can lead to "deeper thinking," with lower and delayed entropy peaks and lower final entropy.

Fig. 1(b, brown line) shows that the above two observations also hold for an RL-trained Video-R1-7B model (Feng et al., 2025). This model has a slightly lower and much later entropy peak than the Qwen2.5-VL-7B-Instruct baseline model, which was used to train Video-R1-7B, and the final entropy is very close to that of the significantly larger Qwen2.5-VL-72B-Instruct model.

Building on these observations, we ask if deeper thinking can be induced in the baseline models directly at inference time, without any training. Specifically, can we manipulate the micro-exploration and micro-exploitation behavior of the baseline instruction-tuned models to enhance their thinking capabilities in a *training-free* manner? To this end, we propose V-Reason, which introduces a small, trainable controller to the LMM value cache, which is adapted only at inference-time. This adaptation involves a few optimization steps of an objective based purely on entropy, without requiring any supervision from data or RL. Instead, the objective encourages more pronounced cycles of micro-exploration and micro-exploitation, by inducing the model to more strongly increase/decrease entropy during these cycles, followed by a final entropy minimization phase. This process prevents entropy from rising too fast during macro-exploration and enables the model to achieve a lower final entropy during macro-exploitation, thus making the baseline model behave more like a stronger reasoning model (see Fig. 1(a,b)). To enhance efficiency, we further introduce a "lite" variant, V-Reason(Lite), which reduces memory and computational overhead by evicting 50% of the lowest-norm video tokens from the KV-cache.

Our results suggest that V-Reason and V-Reason(Lite) bridge the gap between baseline instruction tuned models and RL-trained models in terms of accuracy (see Fig. 1(c)). Notably, our training-free approach mainly guides the baseline model's search for reasoning traces in a more

grounded and controlled manner, avoiding unbounded entropy increases and enabling deeper thinking. It is important to note though that if a base model lacks the knowledge to solve a certain problem, our training-free search-based approach cannot compensate for that limitation. In other words, if the solution lies outside the search space of the model's knowledge, a search-based algorithm cannot discover it. For such problems, training-based approaches would be better suited. While we do not see such limitation often for many video reasoning tasks, we will discuss one such instance in detail in the Section 4 (e.g., the VSI(MRA) task in Fig. 1(c)). Finally, because we have a dedicated entropy minimization phase, we also converge to the final solution trajectory significantly faster, thus producing considerably fewer output tokens on average compared to the RL models (see Fig. 1(d)) which also helps the inference times. Thus, `V-Reason` and `V-Reason(Lite)` bridge the gap with the RL-trained model while producing significantly fewer output tokens. In summary, the paper makes the following **key contributions**:

1. To our knowledge, the problem of *inducing video reasoning without training* has not been previously addressed in the literature. We are the first to introduce a training-free, purely inference-time optimization method to improve video reasoning without SFT or RL.
2. We hypothesize that deeper thinking can be achieved by pronounced micro-exploration and micro-exploitation cycles of the baseline instruction-tuned models and propose `V-Reason` to achieve this. We also provide simple theoretical results for our method.
3. We show that `V-Reason` induces a lower and delayed entropy peak during macro-exploration and a lower final entropy during macro-exploitation, similar to the patterns observed for the reasoning models trained by RL or SFT (see Fig. 1(b)).
4. Extensive experiments on six video reasoning benchmarks show that `V-Reason` achieves an average improvement of **1.4%** over the base model, narrowing the gap to within **0.6%** of the RL-trained Video-R1-7B model (see Fig. 1(c)). We further show gains across model sizes ranging from 3B to 32B and even up to 72B LMMs. We also demonstrate that `V-Reason` is robust/complementary to multiple SOTA decoding methods and perform many ablations.
5. Finally, we show that inference time optimization can lead to more efficient reasoning by significantly reducing the total number of reasoning tokens generated (see Fig. 1(d)). `V-Reason` produces **21.4%** fewer tokens than the base Qwen2.5-7B-Instruct model, and **58.6%** fewer tokens than the RL-trained Video-R1-7B model. This means that its inference time is competitive to the base model and up to **37%** lower than Video-R1-7B on average.

## 2 RELATED WORK

**Reasoning in Large Language Models.** Reasoning in LLMs can be achieved by chain-of-thought prompting, instruction-tuning with CoTs, or reward-based fine-tuning with RL. Existing work on prompting primarily relies on eliciting better CoT reasoning paths from the model (Kojima et al., 2022; Yasunaga et al., 2023; Zhou et al., 2023a). While these methods have achieved high accuracies, few-shot prompting techniques are task-specific, less generalizable and require manual prompt designs for each task. Better prompting techniques require extensive prompt engineering and result in inconsistent performances (Zhou et al., 2023b). Overall, prompting techniques are limited by model-specific and task-specific tuning (Yang et al., 2024b) making them less favorable. Recent works endeavor to improve the CoT prompting by verification (Golovneva et al., 2023) that verifies and controls the intermediate steps generated by the model. Such methods still require CoT prompting and are computationally intensive due to the additional verification steps involved.

Instruction-tuning and reward-based fine-tuning are alternative ways to elicit reasoning in LLMs when additional compute is available for supervision (Magister et al., 2023; Huang et al., 2023; Chung et al., 2022). However, these techniques require supervised CoT data and expensive RL stages to make the model compliant to produce the reasoning or thinking process in specified formats for easy extraction of the answers. Different from the above methods, we seek an efficient framework to enhance reasoning in LMMs via inference-time optimization without any supervised data or training.

**Video Reasoning.** Video Reasoning methods have been introduced recently (Feng et al., 2025; Chen et al., 2025) inspired by the success of LLM reasoning. Video-R1 (Feng et al., 2025) introduces a temporal GRPO loss to specifically improve temporal reasoning capabilities along with a new dataset for training. VideoChat-R1 (Li et al., 2025) introduces a chat model with spatio-temporal reasoning abilities by training with GRPO and rule-based rewards. TinyLLaVA (Zhang et al., 2025a) shows that reasoning can be effective even for smaller models, using a Qwen-3B-VL model trained with standard GRPO and RL-based reward losses. All of the above methods rely on expensive

training to elicit reasoning in LMMs for videos; for instance, training TinyLLaVA on 50K samples takes ~3 days on 4 A100 GPUs, and the cost scales prohibitively for larger models (7B, 32B). To overcome this, we propose an efficient framework that leverages inference-time optimization to enhance the pretrained reasoning abilities of LMMs, achieving higher accuracy with fewer output tokens compared to RL-trained models.

**Inference-time Reasoning Methods.** Inference-time optimization methods (Chefer et al., 2023; Rout et al., 2025) have gained popularity in diffusion models for improving control and consistency. Recent works have explored eliciting reasoning capabilities from LLMs at inference time (Wang & Zhou, 2024; Fu et al., 2025), aiming to reduce computational cost and improve interpretability. Decoding strategies such as CoT-Decoding (Wang & Zhou, 2024) modifies token selection to surface latent reasoning traces, while ThinkLogit (Zhang et al., 2025c) manipulates logits with guidance from a smaller preference model to induce longer reasoning chains. In parallel, sampling-based methods such as min-p (Nguyen et al., 2024) and the concurrent approach top-h (Baghaei Potraghloo et al., 2025) restrict candidate tokens based on probability thresholds or rank cutoffs, improving fluency but without explicitly targeting reasoning. Our method is orthogonal to these approaches: rather than filtering outputs, we optimize the model's intrinsic token distributions during inference and show consistent improvements even when combined with min-p and top-h sampling-based methods.

Other line of works utilize steering to modify the model's behavior for reasoning tasks (Azizi et al., 2025; Belitsky et al., 2025). (Azizi et al., 2025) modifies the hidden states of the model to compress CoT traces by relying on a reasoning-trained model to distinguish concise from verbose reasoning. KV Cache Steering (Belitsky et al., 2025) presents a one-shot intervention in the key-value cache to induce reasoning in small LLMs with steering vectors derived from GPT-4o (Hurst et al., 2024). In contrast to these works that have indirect reliance on a *reasoning-trained* model, we propose an inference-optimization technique that modulates the value-cache to enhance reasoning using only the entropy of the model's output as objective *without any reliance on external model or data*.

## 3 PROPOSED APPROACH: V-REASON

In this section, we describe the proposed `V-Reason`, its inference-time optimization objectives, and supporting theoretical results. We then address practical aspects, including redundancy reduction in video tokens to lower memory costs, and introduce `V-Reason(Lite)` for improved efficiency.

### 3.1 INFERENCE-TIME OPTIMIZATION

Modifying the reasoning behavior of a pre-trained LMM requires two components: a set of reasoning inducing parameters, which are modified or added to the model to improve reasoning, and an optimization objective, to optimize those parameters. As discussed in Section 1, the key goals for `V-Reason` are to: (a) decrease the rate of growth of the output distribution entropy during macro-exploration, by *controlling* the model behavior so as to promote more pronounced cycles of micro-exploration and micro-exploitation during the output generation, and (b) reduce the final entropy during macro-exploitation. To accomplish these objectives, we propose a value-cache controller and a novel inference-time optimization objective.

**Reasoning Inducing Parameters.** We propose to augment the model with the *Value-Cache Controller* shown in Fig. 2(a). This controller, denoted as $\Delta V$, is a small, trainable parameter added to the value cache $\mathbb{V}_L$ of the *last* decoder layer of the model, specifically at the video token locations. All other model layers remain frozen and no modifications are applied to the input or output text tokens. The controller $\Delta V$ is initialized to zero and updated at every $k^{th}$ generated output token ($k > 1$) via the inference-time optimization method discussed below. Note that no optimization is performed for the first token, as that is when the KV-Cache prefilling happens for all layers. To prevent the controller from destabilizing the pretrained model, we introduce the normalization

$$\mathbb{V}_L^{new} = \frac{\mathbb{V}_L + \Delta V}{||\mathbb{V}_L + \Delta V||} \cdot ||\mathbb{V}_L||. \tag{1}$$

This normalization preserves the original magnitude $||\mathbb{V}_L||$ of the cache vector, ensuring that the controller $\Delta V$ introduces only a directional update. This helps maintain a stable forward pass, ensuring consistent output token generation. This normalization is inspired by well-known methods like Weight Normalization (Salimans & Kingma, 2016; Srebro & Shraibman, 2005), which have been shown to have good optimization properties and are beneficial for recurrent and generative models.

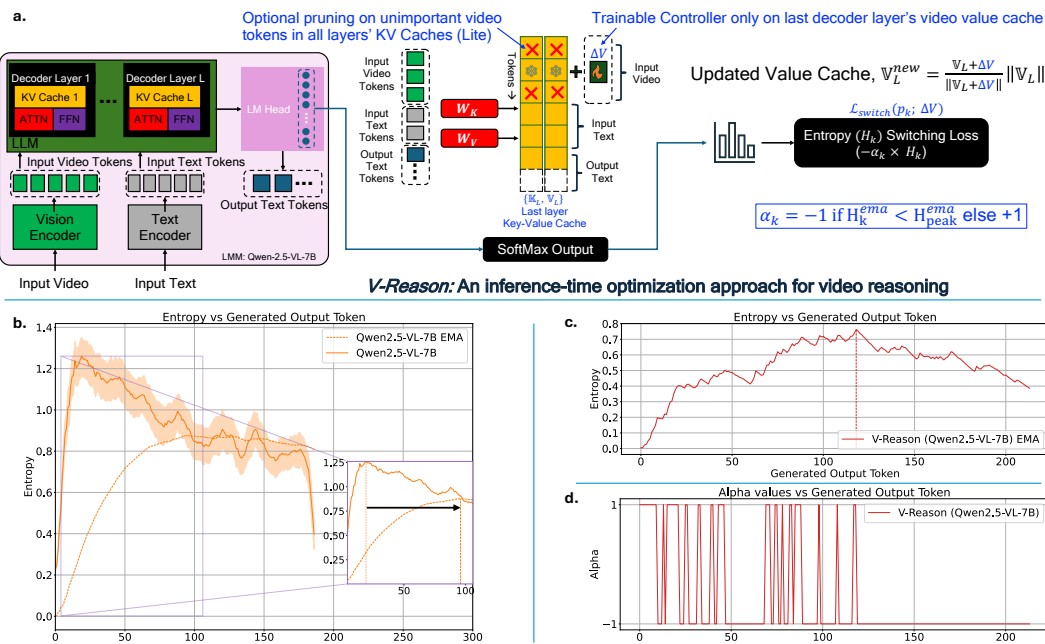

Figure 2: (a) Proposed approach for enhancing video reasoning in a training-free manner using entropy-based objective. V-Reason uses an inference optimization method to modulate the values cache of the last decoder layer with an entropy switching loss ($\mathcal{L}_{switch}$) to further enhance the video reasoning performance. (b) The average entropy plot for Qwen-2.5-VL-7B on the MMVU dataset along with its EMA. The inset depicts the shift in the entropy maxima for the EMA curve denoted by the black arrow (c) EMA entropy plot of V-Reason for a single sample that shows the micro-exploration and micro-exploitation within the macro-exploration phase before the entropy maxima and macro-exploitation phase after. (d) Plot showing the $\alpha_k$ switching in V-Reason for the corresponding example in (c) that ensures bounded entropy updates without a rapid increase.

**Optimization Objective.** In Section 1 and Fig. 1(a), we suggested that the effectiveness of a reasoning model is related to the entropy of its output token distribution. While all reasoning models exhibit a period of macro-exploration, where entropy increases, and macro-exploitation, where it decreases, better models have a macro-exploration stage characterized by lower and delayed entropy maxima. We further posited that this is largely driven by cycles of micro-exploration and micro-exploitation, which prevent the entropy from increasing or decreasing too rapidly. We interpret these cycles as periods where the model temporarily increases the output entropy (exploration) to allow alternative reasoning paths, needed to escape from a current unpromising path. The model then pursues a new path in more detail (exploitation), leading to a decrease of entropy and the potential realization that this new path is itself not promising. The cycle is then repeated. We hypothesize that stronger reasoning models are more decisive in their patterns of micro-exploration and exploitation, which leads to more and/or stronger cycles, thus reducing the rate of *macro* entropy increase. This leads to lower and delayed entropy peaks. It follows that the reasoning power of a model should increase if the model is encouraged to have more *vigorous* micro-exploration/exploitation cycles. After reaching the entropy peak of macro-exploration, the model switches to macro-exploitation, where it pursues a reasoning path in detail to produce an answer, which leads to a decrease of the output entropy. Better models reach lower entropy values at the end of this stage. In this work, we propose to reinforce this behavior by optimizing the value cache controller $\Delta V$ with the *Entropy Switching Loss*:

$$\mathcal{L}_{switch}(\Delta V) \ = -\alpha_k H_k = \alpha_k \sum_{i \in |\mathcal{V}|} p_k^i(\Delta V)\log(p_k^i(\Delta V)) \tag{2}$$

where $p_k$ is the output distribution (softmax after the LM-Head) for every $k^{th}$ token generated ($k > 1$), $H_k$ the entropy of this distribution, and $\alpha_k \in \{-1, +1\}$ is a coefficient that switches between $-1$ and $+1$. The minimization of this loss encourages an increase in the entropy (micro-exploration) when $\alpha_k = 1$ and a decrease (micro-exploitation) when $\alpha_k = -1$. Hence, setting $\alpha_k = 1$ ($\alpha_k = -1$) during the micro-exploration (micro-exploitation) periods, encourages the model to be more decisive in its micro-exploration/exploitation cycles. It is also possible to explore other behaviors, e.g., using this procedure to reinforce micro-cycles during macro-exploration, followed by minimizing entropy alone ($\alpha_k = -1$) during macro-exploitation.

To implement this, we first compute the exponential moving average (EMA) of the entropy at each generation step $t$ (different from $k$, which is the optimization step for the value-cache controller)

$$H_t^{ema} = \beta H_{t-1} + (1 - \beta)H_t \tag{3}$$

where $t > 1$, $\beta$ is a smoothing coefficient (set to 0.98), and $H_0$ is the entropy of the first token which is a small value[1]. The EMA is a low-pass filtered version of the raw entropy, and thus much less noisy, as shown in Fig. 2 (b). It achieves a good trade-off between oscillating too much, due to noise, and switching between increasing and decreasing entropy during micro-cycles, as shown in Fig. 2 (c). Also, because it grows much slower than the raw entropy, following the EMA naturally leads to a lower and delayed entropy peak, as shown in Fig. 2 (b). The switching coefficient $\alpha_k$ is then defined to follow the EMA,

$$\alpha_k = \begin{cases} +1 \text{ if} H_k^{ema} \geq H_{peak}^{ema} \\ -1 \text{ if } H_k^{ema} < H_{peak}^{ema} \end{cases} \tag{4}$$

where, $H_k^{ema}$ is the EMA at the current step, and $H_{peak}^{ema}$ the maximum value of EMA observed before step $k$. This is illustrated in Figure 2 (d). It encourages the entropy to (*i*) increase when the EMA is larger than the last peak, i.e., the EMA is increasing, and to (*ii*) decrease otherwise, i.e., the EMA is decreasing, therefore reinforcing the natural micro-cycles of the model. Once the EMA reaches a global maximum, $\alpha_k$ becomes $-1$ and macro-exploitation begins. This global maximum of entropy can also be seen as a more formal definition of the end of the "thinking" process. A detailed description of the full algorithm is given in Algorithm 1.

Fig. 2 (c) shows the EMA entropy plot of `V-Reason` for a single sample. It is clear that there are more and stronger local minima and maxima depicting the micro-exploration/exploitation cycles before the entropy maxima. This slows the entropy growth during macro-exploration, leading to a delayed peak and substantially more exploration than by the original model. Once the global maximum of the EMA is reached, $\alpha_k$ becomes $-1$ and the model enters the macro-exploitation stage, where it is encouraged to decrease entropy until it arrives at a solution. Overall, the optimization promotes 1) more and/or longer cycles of micro-exploration and micro-exploitation during the macro-exploration stage, which lead to "deeper thinking," with lower and delayed entropy peaks, and 2) a stronger emphasis on entropy minimization during the macro-exploitation stage, which leads to faster convergence to a lower final entropy.

We observe that the optimization of `V-Reason` induces the model to arrive at the final solution significantly faster than CoT-SFT and RL models, which often produce verbose outputs. This can be seen in Fig. 1 (d). Since computation is tied to the length of the output sequence, this also results in significantly more efficient inference than those models. Hence, despite the extra computation needed for the optimization, `V-Reason` has more efficient inference overall (section 3.2). Finally, since `V-Reason` exploits the natural variation in entropy, it adaptively determines how much exploration and exploitation is required by each sample. This makes it robust and adaptable to various datasets and types of video reasoning problems (see Section 4).

**Theoretical Guarantees.** We provide theoretical guarantees that the entropy updates induced by our Entropy Switching Loss remain stable and that our EMA-based objective bounds the oscillations in entropy. The formal statements are below, with assumptions and proofs discussed in Appendix A.

**Proposition 1** (Bounded entropy updates). *Under mild smoothness and boundedness assumptions, one gradient step of size $\eta$ on the Entropy Switching Loss changes entropy by at most*

$$|H_{t+1} - H_t| \leq \eta C + o(\eta),$$

*and the process $\{H_t\}$ remains within the compact interval $[0, \log n]$.*

**Proposition 2** (EMA smoothing bounds oscillations). *For $\beta \in (0, 1)$ close to 1, the EMA acts as a low-pass filter:* (*i*) *it attenuates high-frequency fluctuations of $H_t$,* (*ii*) *delays the attainment of entropy maxima, and* (*iii*) *enforces bounded oscillations by switching $\alpha_k$ to $-1$ once a new global EMA maximum is reached.*

*Proofs*: Please see Appendix A for the proofs of both propositions.

---

[1]The baseline instruction-tuned models are certain about the very first predicted token; it is usually just the `<think>` token, even without RL or CoT-SFT, because of the instruction we give to the model.

Table 1: Comparison of performance of different models on video reasoning benchmarks. #F denotes the number of frames and px denotes the maximum video pixels used, px$\times 28 \times 28$.

| Model | #F/px | VSI-Bench (Acc/MRA) (Yang et al., 2025) | VideoMMMU (Hu et al., 2025) | MMVU (mc) (Zhao et al., 2025) | MVBench (Li et al., 2024b) | TempCompass (Liu et al., 2024) | VideoMME (wo sub) (Fu et al., 2024) | Avg | Avg (wo mra) |
|---|---|---|---|---|---|---|---|---|---|
| GPT-4o (Hurst et al., 2024) | – | 34.0 | 61.2 | 75.4 | – | – | 71.9 | – | – |
| LLaMA-VID (Li et al., 2023) | – | – | – | – | 41.9 | 45.6 | – | – | – |
| VideoLLaMA2 (Cheng et al., 2024) | – | – | – | 44.8 | 54.6 | – | 47.9 | – | – |
| LongVA-7B (Zhang et al., 2024) | – | 29.2 | 23.9 | – | – | 56.9 | 52.6 | – | – |
| VILA-1.5-8B (Lin et al., 2023) | – | 28.9 | 20.8 | – | – | 58.8 | – | – | – |
| Video-UTR-7B (Yu et al., 2025) | – | – | – | – | 58.8 | 59.7 | 52.6 | – | – |
| LLaVA-OneV-7B (Li et al., 2024a) | – | 32.4 | 33.8 | 49.2 | 56.7 | – | 58.2 | – | – |
| Qwen2.5-VL-3B (Bai et al., 2025) | 32/128 | 24.3 (31.6/17.0) | 32.3 | 49.3 | 52.5 | 28.1 | 48.1 | 37.0 | 40.3 |
| V-Reason-3B (Lite) | 32/128 | 26.3 (32.2/20.4) [+0.6/+3.4] | 33.9 [+1.6] | 50.9 [+1.6] | 53.2 [+0.7] | 29.1 [+1.0] | 49.0 [+0.9] | 38.3 [+1.3] | 41.3 [+1.0] |
| V-Reason-3B | 32/128 | 24.7 (31.9/17.5) [+0.3/+0.5] | 33.2 [+0.9] | 50.2 [+0.9] | 52.9 [+0.4] | 30.4 [+2.3] | 48.8 [+0.7] | 37.9 [+0.9] | 41.2 [+0.9] |
| Qwen2.5-VL-7B (Bai et al., 2025) | 16/256 | 26.4 (31.4/21.4) | 47.6 | 59.5 | 60.4 | 72.2 | 50.5 | 49.0 | 53.6 |
| V-Reason-7B (Lite) | 16/256 | 27.9 (34.1/21.6) [+2.7/+0.2] | 47.6 [+0.0] | 63.4 [+3.9] | 60.8 [+0.4] | 71.6 [-0.6] | 51.1 [+0.6] | 49.9 [+0.9] | 54.6 [+1.0] |
| V-Reason-7B | 16/256 | 28.5 (34.5/22.6) [+3.1/+1.2] | 47.8 [+0.2] | 62.2 [+2.7] | 61.0 [+0.6] | 72.3 [+0.1] | 51.1 [+0.6] | 50.2 [+1.2] | 54.8 [+1.2] |
| Video-R1-7B (Feng et al., 2025) | 16/256 | 33.8 (30.5/37.0) | 47.8 | 64.2 | 63.9 | 72.2 | 57.2 | 53.3 | 56.0 |
| Qwen2.5-VL-7B (Bai et al., 2025) | 32/128 | 28.1 (33.8/22.3) | 45.8 | 61.3 | 60.7 | 72.4 | 53.7 | 50.0 | 54.6 |
| V-Reason-7B (Lite) | 32/128 | 30.5 (37.3/23.7) [+3.5/+1.4] | 47.4 [+1.6] | 65.0 [+3.7] | 60.6 [-0.1] | 72.4 [+0.0] | 53.5 [-0.2] | 51.4 [+1.4] | 56.0 [+1.4] |
| V-Reason-7B | 32/128 | 30.3 (37.1/23.4) [+3.3/+1.1] | 46.3 [+0.5] | 62.7 [+1.4] | 60.9 [+0.2] | 73.3 [+0.9] | 54.9 [+1.2] | 51.2 [+1.2] | 55.9 [+1.3] |
| Video-R1-7B (Feng et al., 2025) | 32/128 | 35.6 (30.9/39.2) | 48.8 | 64.0 | 64.1 | 73.3 | 58.7 | 54.1 | 56.6 |

## 3.2 EFFICIENCY CONSIDERATIONS: V-REASON(LITE)

Video reasoning and vision-language LMMs can have high GPU memory costs due to a large number of input video tokens. Adding inference-time optimization to these models at first sight can seem inefficient, as it can further increase inference costs. However, V-Reason has several properties that counteract this hypothesis. First, the controller is only added to the decoder cache of the last model layer. This significantly reduces the memory overhead of storing activations for backpropagation, which reduces to the trainable controller $\Delta V$ and a few feature maps (last decoder layer's value cache, attention output, feedforward layers, and LM-Head). Second, and most important, because V-Reason usually arrives at the final solution with significantly less tokens as shown in Fig. 1(d), both its inference time and computation are much lower than models trained to think.

Nevertheless, we explore an additional avenue for efficiency. Before performing the V-Reason optimization, we *optionally* prune 50% of the video tokens from the KV-Cache of all decoder layers, a variant we refer to as V-Reason(Lite). This significantly reduces the KV-Cache overhead and also halves the size of the trainable controller. Interestingly, we found that for some datasets this also slightly improves V-Reason reasoning performance (perhaps by reducing noise due to unimportant video tokens). To prune out unimportant video tokens, we measure the mean value of the $l_2$ norm of video tokens across all value caches and eliminate the lowest 50% video tokens from both Key and Value Caches of all decoder layers. The trainable controller is then only added to the remaining video tokens in the last decoder layer. The new value update is $\mathbb{V}_L^{new} = \frac{\mathbb{V}_L^{pruned}+\Delta V}{||\mathbb{V}_L^{pruned}+\Delta V||} \cdot ||\mathbb{V}_L||$, which still maintains the magnitude of the unpruned video value cache from equation 1. We empirically find that this reduces the error due to pruning and enables the V-Reason(Lite) models to achieve much higher accuracies than when the value cache norm is altered. Algorithm 2 in Appendix provides the pseudo-code for the lite variant.

## 4 EXPERIMENTS

**Implementation Details.** All experiments use pytorch version 2.5.1+cu121, transformers version 4.52.4, and a single NVIDIA-A100 GPU. Following (Feng et al., 2025), we use multinomial sampling with (temperature=0.1, top-p=0.001) for our experiments unless otherwise noted. See Appendix B for more details.

**Video Reasoning.** We evaluate V-Reason on the Qwen-2.5-VL-Instruct (Bai et al., 2025) model series under 16/32 frames settings (from (Feng et al., 2025)) and maximum video pixels px$\times 28 \times 28$ with px=256/128, respectively. Similar to (Feng et al., 2025), V-Reason is evaluated across 6 video reasoning benchmarks, covering two tasks, Multiple-Choice QA and Regression, evaluated by classification accuracy and Mean Relative Accuracy (MRA) respectively. We report the average accuracy with and without MRA to illustrate the model's performance across different task formulations.

### 4.1 VIDEO REASONING BENCHMARK RESULTS

Table 1 presents a comparison of V-Reason with Qwen2.5-VL-Instruct baselines and the RL-trained Video-R1-7B across multiple video reasoning benchmarks. Green brackets show the gain

of the `V-Reason` model over the baseline, with negative gains in red. Both (at least one) versions of `V-Reason` improve the baseline performance for 15/18 (18/18) model/dataset combinations. Furthermore, the gain is of at least **1.5 points** for 12/18 combinations and can be as high as **3.9 points**. In many cases, these gains are a substantial part of the gap between the baseline and the RL-trained model. For example, for MMVU and 7B-256px models the 63.4 point accuracy of `V-Reason` (Lite) brings the relatively poor 59.5 point baseline close to the 64.2 point accuracy of the Video-R1. For the 128 px model, `V-Reason` even surpasses Video-R1 (**65.0 vs. 64.0**). This model also matches Video-R1 on TempCompass (73.3 each), and nearly closes the gap on VideoMMMU (47.4 vs. 48.8). These very significant gains show that the baseline model already has a significant ability to reason, which RL brings to the surface, but can also be mostly unlocked by much less expensive inference time optimization of `V-Reason`. The only tasks where RL optimization proves particularly effective are the regression-style tasks (e.g., VSI-Bench), which are probably underrepresented in pretraining, as can be seen by the very poor baseline performance. Here, Video-R1 exhibits stronger performance (MRA 39.2 vs. 23.7 for `V-Reason`). Since `V-Reason` relies on the pretrained model knowledge to explore alternative paths during reasoning, it cannot cover the gap on these. Nevertheless, `V-Reason` obtains a **+1.4%** improvement over the baseline.

Overall, across model scales, input resolutions, and tasks other than regression (without MRA), `V-Reason` has average gains in **[+0.9,1.3]%** over Qwen2.5-VL, at the 3B scale. At the 7B scale, `V-Reason`/`V-Reason(Lite)` reaches 54.8%/54.6% (256 px) and 55.9%/56.0% (128 px) average accuracy, narrowing the gap to Video-R1-7B (56.6% at 128 px) to within **0.6%**.

**Impact of Frames and Resolution.** `V-Reason` is robust to different frame counts and resolutions, making it adaptable to resource-constrained settings. For a comparable compute budget (256 px/16 frames vs. 128 px/32 frames), accuracy improves with more frames. Consistent with this trend, `V-Reason` shows larger average gains at 32 frames (**+1.4%**) compared to 16 frames (**+1.0%**).

**Full vs. Lite.** Both Full and Lite `V-Reason` variants surpass the base model, with Lite performing comparably or slightly better in several cases (e.g., **+1.0%** at 3B and **+1.4%** at 7B for 128 px). The Full variant offers a small **+0.2%** improvement at 256 px with 16 frames, suggesting that structural pruning and reasoning enhancements provide complementary benefits.

**Output Sequence Length.** As shown in Fig. 1(d), `V-Reason` substantially reduces output token length (**58.6% reduction** over Video-R1). Table 5 (Appendix C) shows that this translates into significant latency savings of up to **67%** (**37%** on average). The fact that this holds for both Full and Lite variants indicates that the gains stem from the proposed optimization rather than pruning alone.

**Larger Models.** To test the scalability of `V-Reason`, we further evaluate it on larger Qwen2.5-VL backbones (32B and 72B) using MMVU. Table 2 shows that `V-Reason` continues to provide significant gains, e.g., **+3.0%** (**72.0 vs. 69.0**) for the 32B model, demonstrating that reasoning benefits compound with scale rather than diminishing. For the 72B model, which is already strong, `V-Reason` still yields a **+0.4% gain** (73.0 vs. 72.6). These results show that `V-Reason` generalizes to larger models and provides benefits even at the frontier of large-scale video reasoning models.

Table 2: Large model results on MMVU.

| Model | MMVU |
|---|---|
| Qwen-2.5-VL-32B | 69.0 |
| V-Reason-32B | **72.0 [+3.0]** |
| Qwen-2.5-VL-72B | 72.6 |
| V-Reason-72B | **73.0 [+0.4]** |

**Comparison with Decoding Methods.** As shown in Table 3, our method is robust and complementary to different decoding strategies with significant improvements over SOTA approaches such as *min-p* (Nguyen et al., 2024) and (concurrent) *top-H* (Baghaei Potraghloo et al., 2025). For the Qwen-2.5-VL-7B model, using the best *min-p* decoding with `V-Reason(Lite)` yields a gain of **+2.0** points on MMVU, while combining with best *top-H* decoding provides a smaller improvement of **+0.4**. On higher temperatures, *min-p* loses significant accuracy but `V-Reason(Lite)` is able to restore it back (**+6.3%**). Most notably, `V-Reason-7B (Lite)` achieves the highest score of **65.0**, corresponding to a further **+2.8** gain over the best decoding baseline.

Table 3: Comparison with Alternative Decoding Methods.

| Qwen-2.5-VL-7B | temp | top-p | MMVU |
|---|---|---|---|
| min-p | 0.3 | 0.9 | 61.8 |
| min-p+V-Reason(Lite) | 0.3 | 0.9 | **63.8 [+2.0]** |
| top-H | 0.3 | 0.9 | 60.2 |
| top-H+V-Reason(Lite) | 0.3 | 0.9 | **61.1 [+0.9]** |
| min-p | 1.0 | 0.9 | 55.0 |
| min-p+V-Reason(Lite) | 1.0 | 0.9 | **61.3 [+6.3]** |
| top-H | 1.0 | 0.9 | 62.2 |
| top-H+V-Reason(Lite) | 1.0 | 0.9 | **62.6 [+0.4]** |
| V-Reason-7B (Lite) | 0.1 | 0.001 | **65.0 [+2.8]** |

**Qualitative Results.** Figure 3 exemplifies the reasoning differences between `V-Reason` and the baseline, also showing their entropy profiles. The entropy plots reveal that `V-Reason` has a delayed

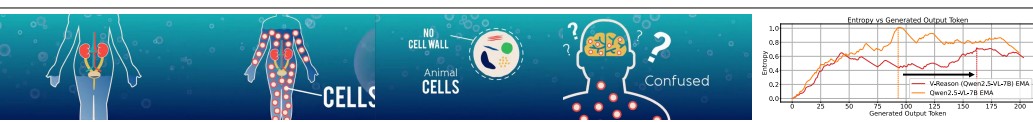

**Question:** When the phenomenon in the video occurs, which of the following options is the most important to adjust? A. Renin-angiotensin system B. Hypothalamic-pituitary-thyroid axis C. Adrenaline and norepinephrine D. Melatonin secretion E. Sodium-potassium pump activity

**Qwen-2.5-VL-7B:**

Let me think through this. The video seems to be discussing a condition where cells swell, which could be related to fluid retention or electrolyte imbalance. Given the options, the sodium-potassium pump activity is crucial for maintaining the balance of sodium and potassium ions across cell membranes, which is essential for cell volume regulation. \n\nThe renin-angiotensin system is involved in blood pressure regulation and fluid balance, but it's more about the overall pressure and not specifically about cell swelling. The hypothalamic-pituitary-thyroid axis is primarily involved in metabolism and energy production, not directly in cell swelling. Adrenaline and norepinephrine are involved in the body's stress response and can affect heart rate and blood pressure, but not directly cell swelling. Melatonin secretion is related to sleep and circadian rhythms, not cell swelling.\n\nTherefore, the most important to adjust in this context would be the sodium-potassium pump activity.\n\n<answer>E</answer>

**V-Reason-7B:**

Let me think through this. The video seems to be discussing a condition where cells swell, which is a common symptom of various medical conditions. The options provided are related to different endocrine systems and their functions. \n\n- The Renin-angiotensin system is involved in blood pressure regulation and fluid balance.\n- The Hypothalamic-pituitary-thyroid axis regulates the thyroid gland and its hormones.\n- Adrenaline and norepinephrine are part of the sympathetic nervous system, involved in the \"fight or flight\" response.\n- Melatonin is a hormone that regulates sleep and wake cycles.\n- The Sodium-potassium pump activity is crucial for maintaining the electrical potential across cell membranes.\n\nGiven that the video is about cell swelling, which can be a result of fluid retention or imbalances in electrolyte levels, the most relevant system to adjust would be the Renin-angiotensin system, as it directly affects fluid balance and blood pressure.\n\n<answer>A</answer>

Figure 3: Qualitative result: An example output and comparison with the baseline Qwen-2.5-VL-7B together with its entropy plot shown on the top right. The black arrow in the entropy plot denotes the shift in the EMA peak demonstrating longer exploration for V-Reason compared to the baseline. See other results in H.

EMA peak and a lower overall entropy, encouraging extended exploration that ultimately enables the model to reach the correct solution. As highlighted in red, the baseline initially follows a promising trajectory but subsequently diverges onto an incorrect reasoning path, which leads to the wrong answer. In contrast, V-Reason identifies an alternative path precisely at the point where the baseline falters, and this revised trajectory, shown in green, successfully leads to the correct answer. Please see Appendix H for other examples.

**Alternative Losses.** The switching loss in equation 2 supports various behaviors beyond that encouraged by V-Reason. Two extreme alternatives are enforcing strictly increasing entropy (max-entropy, $\alpha_k = 1, \forall k$) and strictly decreasing entropy (min-entropy, $\alpha_k = -1, \forall k$). Table 4 shows both approaches are clearly inferior to V-Reason. However, it is interesting to note that even these basic strategies (encourage macro-exploration or macro-exploitation only) improve on the performance of the baseline model. This confirms the importance of the output distribution entropy on the reasoning ability of LMMs.

Table 4: Optimization objective ablations.

| Method | MMVU |
|---|---|
| Qwen-2.5-VL-7B | 61.3 |
| Min-Entropy (Lite) | 62.1 [+0.8] |
| Max-Entropy (Lite) | 63.8 [+2.5] |
| V-Reason(Lite) | **65.0 [+3.7]** |

**Optimization Step-size.** As shown in Figure 4 (Appendix E), V-Reason consistently outperforms the base model across different step-sizes $k$, highlighting a trade-off between accuracy (better with smaller step-sizes) and efficiency (faster with fewer steps). More analyses and ablations are provided in Appendix D and E. Please see Appendix F for a discussion on the limitations of V-Reason and the future work G.

## 5 CONCLUSION

In this paper, we introduced V-Reason, a novel training-free framework designed to enhance reasoning in videos, along with a value-cache controller that enables inference-time optimization. Our method leverages a theoretically-grounded entropy-based objective to reinforce the micro-exploration and micro-exploitation behaviors observed across models. This design effectively mitigates unbounded entropy growth during early generation steps, resulting in lower final entropy, a characteristic of stronger models. We further proposed V-Reason(Lite), a "Lite" variant which improves the memory by pruning low $l_2$-norm entries in the value cache. Extensive experiments across multiple benchmarks demonstrate that V-Reason narrows the gap to RL–trained models (e.g., Video-R1) to within **0.6%**, while substantially reducing output token length (↓**58.6%**); this also results in lower (↓**37%**) inference time than Video-R1. Moreover, V-Reason consistently improves performance across model scales ranging from 3B to 72B parameters and remains robust to variations in frame sampling, pixel resolution, decoding techniques, and other hyperparameter configurations.

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

## A  THEORETICAL ANALYSIS: BOUNDING ENTROPY UNDER SWITCHING LOSS

Let the vocabulary size be $n = |\mathcal{V}|$. At generation step $t$, the model (with value-cache controller parameters $\Delta V$) produces logits $z_t \in \mathbb{R}^n$ and probabilities

$$p_t(\Delta V) = \text{softmax}(z_t(\Delta V)), \qquad \sum_i p_t^i = 1.$$

The Shannon entropy of this distribution is

$$H_t(\Delta V) := -\sum_{i=1}^{n} p_t^i(\Delta V) \log p_t^i(\Delta V),$$

and its exponential moving average (EMA) is

$$H_t^{ema} = \beta H_{t-1}^{ema} + (1 - \beta) H_t, \quad \beta \in (0, 1).$$

The Entropy Switching Loss at optimization step $k$ is

$$\mathcal{L}_{switch}(\Delta V) = -\alpha_k H_k(\Delta V),$$

where the coefficient $\alpha_k \in \{-1, +1\}$ is defined as

$$\alpha_k = \begin{cases} +1 & \text{if } H_k^{ema} \geq H_{peak}^{ema}, \\ -1 & \text{otherwise}, \end{cases}$$

with $H_{peak}^{ema}$ denoting the maximum EMA value observed before step $k$.

**Assumptions.**  We make the following assumptions:

1. Logits $z_t(\Delta V)$ are smooth in $\Delta V$, and $\partial z_t / \partial \Delta V$ is bounded. From equation 1, $\mathbb{V}_L^{new} = \frac{\mathbb{V}_L + \Delta V}{||\mathbb{V}_L + \Delta V||} \cdot ||\mathbb{V}_L||$. So, $\partial z_t / \partial \Delta V$ being bounded is a valid assumption because the update to value cache is bounded by the normalization factor which only provides a directional update.

2. The optimizer uses a bounded step size (learning rate) $\eta > 0$ and updates are sufficiently small per step (i.e., standard stochastic gradient/Lipschitz assumptions).

3. Vocabulary size is finite, hence $H_t \in [0, \log n]$ for all $t$.

**Preliminaries.**  Differentiating the entropy with respect to logits yields

$$\nabla_z H = -J_p^\top (\mathbf{1} + \log p),$$

where $J_p = \partial p / \partial z$ is the softmax Jacobian. Since $\|J_p\|$ is bounded and $\mathbf{1} + \log p$ is finite (as $p_i \in (0, 1]$), we obtain

$$\|\nabla_{\Delta V} H\| \leq C$$

for some constant $C$.

**Proposition 1** (Bounded entropy updates)**.** *Under the assumptions above, one gradient step of size $\eta$ on $\mathcal{L}_{switch}$ changes entropy by at most*

$$|H_{t+1} - H_t| \leq \eta C + o(\eta),$$

*and the process $\{H_t\}$ remains in the compact interval $[0, \log n]$. Here, $o(\eta)$ denotes the higher-order terms from the Taylor expansion of $H(\Delta V)$ around the current iterate.*

*Proof.*  First, the gradient of entropy with respect to controller parameters is

$$\nabla_{\Delta V} H_k(\Delta V) = \frac{\partial H_k}{\partial z_k} \frac{\partial z_k}{\partial \Delta V}.$$

**Bounding $\nabla_z H_k$.**  For softmax probabilities bounded away from 0 and 1, the Jacobian $J_p = \partial p_k / \partial z_k$ satisfies $\|J_p\|_2 \leq 1$. Moreover, the entropy gradient w.r.t. logits is

$$\nabla_z H_k = -J_p^\top (\mathbf{1} + \log p_k),$$

and $\|\mathbf{1} + \log p_k\|_2 \leq \sqrt{n} \max_i |1 + \log p_k^i| \leq C_1$ for some constant $C_1$ depending on $n$ and $\epsilon$ (the lower bound on softmax probabilities). Therefore,

$$\|\nabla_z H_k\|_2 \leq C_1.$$

**Bounding $\nabla_{\Delta V} H_k$.** Since $z_k$ is $L_z$-Lipschitz in $\Delta V$,

$$\|\nabla_{\Delta V} H_k\|_2 = \|\nabla_z H_k \cdot \partial z_k / \partial \Delta V\|_2 \leq C_1 L_z := L_H.$$

**Bounding one gradient step.** A single gradient step updates the controller:

$$\Delta V \leftarrow \Delta V + \eta \alpha_k \nabla_{\Delta V} H_k.$$

Using the Lipschitz property of $H_k$ w.r.t $\Delta V$,

$$|H_k(\Delta V + \eta \alpha_k \nabla_{\Delta V} H_k) - H_k(\Delta V)| \leq \eta \|\nabla_{\Delta V} H_k\|_2 \leq \eta L_H.$$

**Global bounds.** Since $H_k \in [0, \log n]$ by definition, this step-size bound guarantees the entropy remains in $[0, \log n]$ after each update.

$\square$

**Proposition 2** (EMA smoothing bounds oscillations). *For $\beta \in (0, 1)$ close to 1, the EMA acts as a low-pass filter:* (i) *it attenuates high-frequency fluctuations of $H_t$,* (ii) *delays the attainment of entropy maxima, and* (iii) *enforces bounded oscillations by switching $\alpha_k$ to $-1$ once a new global EMA maximum is reached.*

*Proof.* (i) The recursion $H_t^{ema} = \beta H_{t-1}^{ema} + (1 - \beta) H_t$ is a causal low-pass filter, suppressing fast oscillations. (ii) Because $H^{ema}$ averages over past values, peaks in $H_t$ appear later and at lower amplitude in $H^{ema}$, creating delayed switching. (iii) Once $H^{ema}$ reaches a global maximum, $\alpha = -1$, turning the loss into an entropy-minimization objective. This guarantees the entropy trajectory descends after each peak, bounding the amplitude of oscillations. $\square$

**Discussion.** The trivial upper bound $H_t \leq \log n$ already prevents unbounded entropy; Proposition 1 strengthens this by showing the optimization dynamics cannot instantaneously jump arbitrarily close to $\log n$ provided the learning rate is small and gradients are bounded. In practice, this prevents pathological "entropy blow-ups" during optimization. EMA smoothing makes the switching decision depend on sustained increases in entropy rather than on single noisy spikes. These results imply that the Entropy Switching Loss enforces *bounded micro-cycles* of exploration and exploitation: entropy increases are promoted only when sustained (captured by $H^{ema}$), while decreases are enforced once a peak is reached. This yields lower and delayed entropy maxima, consistent with the empirical patterns of stronger reasoning models.

Concurrent work, Top-H (Baghaei Potraghloo et al., 2025), formalizes entropy bounds in the decoding step by solving (approximately) an entropy-constrained minimization problem that upper-bounds the randomness of the truncated distribution while keeping divergence from the model distribution small. Our approach uses a complementary perspective: rather than imposing a hard constraint on the sampling distribution at each decoding step, we *optimize the controller* so that the model's intrinsic token distributions themselves enter phases of controlled exploration and exploitation (via maximizing/minimizing $H$ at different times). The EMA-based switching mirrors the time-adaptive, entropy-aware thresholds used in Top-H while operating *inside* the model (controller optimization) rather than as an external truncation rule. Empirically and theoretically, both approaches rely on the same fundamental fact: *entropy is a natural, bounded quantity* that can be used as a control signal to trade-off diversity and consistency in generation.

## B  IMPLEMENTATION DETAILS

**Hyperparameters.** AdamW optimizer is used to update the controller with no weight decay. A step size of $k = 4$ is used as default unless otherwise specified and the best accuracy is reported over a grid search of 10 learning rates from `5e-5` to `5e-4`. The gradient norm of the value-cache controller is clipped to 1.0. We used $\beta = 0.98$ smoothing factor for EMA.

---

**Algorithm 1** Autoregressive LMM inference with `V-Reason`

---

**Require:** Pretrained LLM $f_\theta$; Encoder $\mathcal{E}$; video frames $\mathcal{V}$; text prompt $\mathcal{X}$; Sampler SAMPLE(;); maximum length $L_{\max}$; temperature $\tau$; vocabulary $\mathcal{W}$.
**Ensure:** Generated text $\hat{\mathbf{y}}$.
1: **function** UPDATEV(V)
2: $\quad$ $\mathsf{V_L}' \leftarrow \frac{\mathbb{V}_L + \Delta V}{\|\mathbb{V}_L + \Delta V\|} \|\mathbb{V}_L\|$ $\qquad\qquad\qquad\qquad\qquad$ ▷ add trainable offset and normalize
3: $\quad$ **return** $\mathsf{V_L}'$
4: **end function**
5: **function** OPTIMIZE($\boldsymbol{\ell}_N, \Delta V, k$)
6: $\quad$ $p_k \leftarrow$ SOFTMAX($\boldsymbol{\ell}_N$)
7: $\quad$ $H_k \leftarrow -\sum_{i \in |\mathcal{W}|} p_k^i(\Delta V) \log(p_k^i(\Delta V))$
8: $\quad$ $\alpha_k = \begin{cases} -1 \text{ if } H_k^{ema} < H_{peak}^{ema}, \\ +1 \text{ otherwise,} \end{cases}$ $\qquad\qquad\qquad$ ▷ compute alpha
9: $\quad$ $\mathcal{L}_{switch}(p_k; \Delta V) \leftarrow -\alpha_k H_k$ $\qquad\qquad\qquad\qquad$ ▷ compute loss
10: $\quad$ $\Delta V \leftarrow \arg\min \mathcal{L}_{switch}(p_k; \Delta V)$ $\qquad\qquad\qquad$ ▷ update parameters
11: $\quad$ **return** $\Delta V$
12: **end function**
13: $\mathbf{z}_{1:N} \leftarrow \mathcal{E}(\mathcal{V}, \mathcal{X})$
14: $(\boldsymbol{\ell}_N, \mathsf{KV}) \leftarrow f_\theta(\mathbf{z}_{1:N})$ $\qquad\qquad\qquad\qquad$ ▷ prefill: compute logits and full KV cache
15: $\hat{y}_1 \leftarrow$ SAMPLE($\boldsymbol{\ell}_N, \tau$)
16: $\hat{\mathbf{y}} \leftarrow [\hat{y}_1]$
17: $t \leftarrow 1$
18: **while** $t < L_{\max}$ and $\hat{y}_t \neq$ [EOS] **do**
19: $\quad$ $\mathsf{V} \leftarrow$ UPDATEV($\mathsf{V}, \mathcal{I}_v, \pi$)
20: $\quad$ $\Delta V \leftarrow$ OPTIMIZE($\boldsymbol{\ell}_N, \Delta V, k$)
21: $\quad$ $(\boldsymbol{\ell}_{N+t}, \mathsf{KV}) \leftarrow f_\theta(\hat{y}_t \mid \mathsf{KV})$
22: $\quad$ $\hat{y}_{t+1} \leftarrow$ SAMPLE($\boldsymbol{\ell}_{N+t}, \tau$)
23: $\quad$ $\hat{\mathbf{y}} \leftarrow [\hat{\mathbf{y}}; \hat{y}_{t+1}]$
24: $\quad$ $t \leftarrow t + 1$
25: **end while**
26: **return** $\hat{\mathbf{y}}$

---

**Evaluation.** Classification accuracy is computed as the proportion of correct answers to the multiple-choice QA. Mean Relative Accuracy measures the proportion of predictions whose relative error falls below a series of thresholds ranging from 0.5 to 0.95. The final score is the average accuracy across all thresholds. For VSI-Bench, we report both classification accuracy and MRA individually, as well as their average. To compute the overall average accuracy across all six datasets, we divide by seven, treating the two scores from VSI-Bench separately in addition to the other datasets. When calculating the average accuracy without considering MRA, we divide by six, using only the accuracy score from VSI-Bench along with the scores from the remaining datasets.

## C INFERENCE TIME AND GPU MEMORY

Table 5 presents the inference time, measured in seconds, of the baseline Qwen-2.5-VL-7B, `V-Reason`-7B, `V-Reason`-7B (Lite), and Video-R1 across the six video reasoning benchmarks. All experiments were conducted on input videos with maximum video pixels set to $128 \times 28 \times 28$ and 32 frames temporal length. The reported results are the average over 50 samples.

From the results, it is evident that `V-Reason` and `V-Reason`(Lite) consistently outperforms Video-R1 in terms of wall-clock inference time except for VideoMMMU. Specifically, `V-Reason` reduces inference time by approximately **20–67%** compared to Video-R1 across the evaluated benchmarks. For instance, on TempCompass, the inference time decreases from 11.8 seconds per sample to 3.9 seconds per sample, while on MVBench, the reduction is from 10.7 seconds per sample to 4.1 seconds per sample. Fig. 1(d) shows that `V-Reason` has the maximum average output token count for VideoMMMU dataset and so using a step-size of 4 results in more number of optimization steps as compared to other datasets. This explains the anomaly observed in VideoMMMU results where the inference time is higher than Video-R1-7B. Further, comparing `V-Reason` and `V-Reason`(Lite) shows that token pruning introduces additional latency that increases the inference time marginally (+0.23 seconds) as compared to the full version without any pruning. These results highlight that

---

**Algorithm 2** Autoregressive LMM inference with `V-Reason(Lite)`

---

**Require:** Pretrained LLM $f_\theta$; Encoder $\mathcal{E}$; video frames $\mathcal{V}$; text prompt $\mathcal{X}$; Sampler SAMPLE(;);
     maximum length $L_{\max}$; temperature $\tau$; pruning policy $\pi$ (e.g., keep ratio $r$ by importance).
**Ensure:** Generated text $\hat{\mathbf{y}}$.

1: **function** PRUNEKV(KV, $\mathcal{I}_v$, $\pi$)
2:      $\mathcal{S} \leftarrow \text{Score}(\text{KV}, \mathcal{I}_v)$                                                  ▷ low L2-norm
3:      $\mathcal{K} \leftarrow \text{Select}(\mathcal{I}_v, \mathcal{S}, \pi)$                           ▷ indices to keep among video positions
4:      $\mathcal{M} \leftarrow \{\text{all text positions}\} \cup \mathcal{K}$                                  ▷ full keep-set
5:      $\text{KV}' \leftarrow \text{IndexSelect}(\text{KV}, \mathcal{M})$            ▷ prune keys/values along sequence dimension
6:      **return** $\text{KV}'$
7: **end function**
8: $\mathbf{z}_{1:N} \leftarrow \mathcal{E}(\mathcal{V}, \mathcal{X})$
9: $(\boldsymbol{\ell}_N, \text{KV}) \leftarrow f_\theta(\mathbf{z}_{1:N})$                       ▷ prefill: compute logits and full KV cache
10: $\mathcal{I}_v \leftarrow \{1, \ldots, N_v\}$                                 ▷ positions of video tokens
11: $\text{KV} \leftarrow \text{PRUNEKV}(\text{KV}, \mathcal{I}_v, \pi)$                   ▷ KV-cache pruning for efficiency
12: $\hat{y}_1 \leftarrow \text{SAMPLE}(\boldsymbol{\ell}_N, \tau)$
13: $\hat{\mathbf{y}} \leftarrow [\hat{y}_1]$
14: $t \leftarrow 1$
15: $\hat{\mathbf{y}} \leftarrow \text{AutoRegressive}[\hat{\mathbf{y}}; \hat{y}_1]$         ▷ inference optimization same as **Algorithm 1**
16: **return** $\hat{\mathbf{y}}$

---

`V-Reason`-7B and `V-Reason`-7B (Lite) achieves a significant efficiency advantage in wall-clock inference time over the RL-trained model while narrowing the gap to within 0.6% accuracy as demonstrated in Table 1.

We report the peak GPU memory usage for all models and compare `V-Reason(Lite)` with `V-Reason` to show the benefit of our pruning variant in reducing GPU memory requirements. Table 6 shows that both `V-Reason` and `V-Reason(Lite)` increase the memory overhead slightly compared to the baseline Qwen-2.5-VL-7B and the Video-R1-7B model as expected due to the additional memory overhead in optimization. Note that the memory overhead is much lower than optimizing for all decoder layers in the KV-cache. To further reduce the overhead, we introduced the lite variant `V-Reason(Lite)`. The table shows that `V-Reason(Lite)` reduces the average memory requirement across all datasets by **11.6%** as compared to the full variant. In particular, the memory requirements drop by **20%** on datasets with longer output token count length such as VideoMMMU (see Fig. 1(d)) suggesting the effectiveness of the proposed Lite variant. Notably, the peak GPU memory of `V-Reason(Lite)` method is always lower than 32GB for the 7B model (on the datasets tested). This shows that the proposed lite variant is more suited for relatively smaller GPUs (e.g., 32GB V100 GPUs) and would not require more expensive GPUs like the Full variant.

**Trainable memory computation example for the controller.** Let us assume a fixed video token length of 1920 for analysis. Then the proposed controller introduces a parameter tensor of shape $(1, 4, 1920, 128)$ for Qwen-2.5-VL-7B model, amounting to $N = 983{,}040$ trainable scalars. In FP32, this corresponds to $N \times 4 \,\text{bytes} = 3.84 \,\text{MiB}$ of weights, while in FP16 the footprint is 1.92 MiB. During training with the AdamW optimizer, additional memory is required for the gradient and two moment estimates of the same size as the parameters. Thus, in pure FP32 training the memory becomes $4 \times 3.75 = 15.36 \,\text{MiB}$ (weights + gradients + $m$ + $v$). Since, the controller is used only as an additive bias (element-wise addition) the arithmetic cost is negligible ($\sim N$ adds, i.e., $< 10^6$ adds). The operation above is tiny compared to the bulk of transformer computation (attention and large dense projections), which typically entail orders of magnitude more FLOPs per token for typical hidden sizes and sequence lengths; therefore the controller's compute overhead is minimal in most deployments. Note that the total GPU memory required for inference-time optimization will also include the memory required for storing the activations and gradients of the last decoder layer in the model as discussed above.

## D    ANALYSIS ON VIDEO DURATION

We investigate the effect of video duration on the performance of `V-Reason` using the VideoMME dataset, which provides annotations for short, medium, and long videos. Specifically, short videos

Table 5: Inference time (in seconds/sample) of Qwen-2.5-VL-7B, V-Reason-7B (Lite), V-Reason-7B, and Video-R1-7B across different video reasoning benchmarks. Averaged over 50 samples from each dataset.

| Model | VSI-Bench | VideoMMMU | MMVU | TempCompass | MVBench | VideoMME | Average |
|---|---|---|---|---|---|---|---|
| Qwen-2.5-VL-7B | 3.80 | 9.02 | 6.73 | 2.86 | 3.30 | 4.37 | 5.01 |
| Video-R1-7B | 10.17 | **11.72** | 11.61 | 11.77 | 10.69 | 11.42 | 11.23 |
| V-Reason-7B (Lite) | 5.43 [↓46.6%] | 14.18 [↑21.0%] | 8.86 [↓23.7%] | 4.18 [↓64.5%] | 4.45 [↓58.4%] | 6.64 [↓41.9%] | 7.29 [↓35.1%] |
| V-Reason-7B | 5.06 [↓50.2%] | 13.83 [↑18.0%] | 9.28 [↓20.0%] | 3.87 [↓67.1%] | 4.13 [↓61.4%] | 6.18 [↓45.9%] | 7.06 [↓37.1%] |

Table 6: Peak GPU memory (in GB) of Qwen-2.5-VL-7B, V-Reason-7B (Lite), V-Reason-7B, and Video-R1-7B across different video reasoning benchmarks. Averaged over 50 samples from each dataset.

| Model | VSI-Bench | VideoMMMU | MMVU | TempCompass | MVBench | VideoMME | Average |
|---|---|---|---|---|---|---|---|
| Qwen-2.5-VL-7B | 16.55 | 16.65 | 16.60 | 16.47 | 16.51 | 16.53 | 16.55 |
| Video-R1-7B | 16.70 | 16.74 | 16.73 | 16.68 | 16.70 | 16.69 | 16.71 |
| V-Reason-7B | 23.95 | 38.48 | 29.91 | 22.32 | 23.28 | 25.56 | 27.25 |
| V-Reason-7B (Lite) | 22.41 [↓6.4%] | 30.79 [↓20.0%] | 25.05 [↓16.2%] | 21.45 [↓3.9%] | 21.86 [↓6.1%] | 22.89 [↓10.5%] | 24.08 [↓11.6%] |

are less than two minutes in duration, medium videos range from 4 to 15 minutes, and long videos span 30 to 60 minutes. Table 7 presents a detailed breakdown of the results for both V-Reason and its Lite variant across these duration categories. The full V-Reason model achieves notable gains, with a substantial improvement on short videos (**+1.8%**) and notable gains on medium (**+0.8%**) and long (**+0.9%**) videos. The Lite variant of V-Reason also yields a significant improvement on short videos (**+1.8%**), comparable to the full model, but its performance decreases for medium and long videos. We attribute this decline to pruning, which likely removes important temporal or contextual details, thereby reducing accuracy for longer content.

# E    ABLATION STUDIES

In this section, we present additional ablation studies to assess the impact of the proposed pruning strategy and the hyperparameters used during inference-time optimization, including optimization step-size (update frequency) and learning rate, and we further analyze the frequency of alpha values before entropy maxima.

**Pruning-Only.** Table 8 compares V-Reason to a baseline model that implements pruning only. This shows that it is effective in maintaining the original performance with only **-0.2%** decrease on average across all datasets. Surprisingly, it also has small gains over the baseline on the VSI-Bench and TempCompass datasets. When V-Reason is combined with pruning, the average gain (without MRA) increases from $-0.2$ to **1.3**. This shows that the reasoning gains derive mostly from the inference optimization. Furthermore, Table 8 reports results for V-Reason with a fixed learning rate of 3e-4 across six datasets. The method maintains the average performance reported in Table 1 under this setting with similar gains observed on VSI-Bench, VideoMMMU, and MMVU datasets and only a negligible drop in the performance on MVBench, Tempcompass, and VideoMME datasets, highlighting its robustness to variations in optimization hyperparameters.

**Optimization Step-size.** Figure 4 shows an ablation on optimization step-size on MMVU dataset. It shows that accuracy increases with decreasing step-size. Since smaller step-sizes correspond to more optimization steps, there is a trade-off between efficiency and accuracy (fewer steps lead to faster inference). Notably, V-Reason outperforms the base model for all step-sizes, demonstrating that even a few optimization steps can guide the model towards improved reasoning paths.

**Alpha Switching.** Figure 5 shows the histogram of alpha values before the EMA peak is attained. V-Reason sacrifices a few micro-exploration steps ($\alpha = 1$) for a substantially larger number of micro-exploitation steps ($\alpha = -1$), suggesting that it pursues more alternative paths during macro-exploration. This lengthens the macro exploration stage and delays the overall entropy peak.

# F    LIMITATIONS

Although V-Reason demonstrates consistent improvements across benchmarks, there are certain limitations. First, our approach relies on the knowledge of the pretrained model to explore alternative paths during the thinking process and so for certain tasks that are poorly represented in the pretrained model, V-Reason cannot fully bridge the gap to training-based approaches. For example, on the regression task on VSI-Bench dataset V-Reason obtains only a modest improvement of **+1.4%**

Table 7: Comparison of Qwen-2.5-VL-7B, V-Reason, and V-Reason-7B (Lite) on VideoMME dataset. The differences with the baseline are denoted in red and green colors.

| Model | Mean Acc. | Short | Medium | Long |
|---|---|---|---|---|
| Qwen-2.5-VL-7B | 53.7 | 64.6 | 50.4 | 46.1 |
| V-Reason-7B (Lite) | 53.5 [–0.2] | **66.4 [+1.8]** | 49.7 [–0.8] | 44.3 [–1.8] |
| V-Reason-7B | **54.9 [+1.2]** | **66.4 [+1.8]** | **51.2 [+0.8]** | **47.0 [+0.9]** |

Table 8: Ablation studies on pruning and learning rates for the variant using 128 px and 32 frames.

| Model | VSI-Bench (Acc/MRA) | VideoMMMU | MMVU (mc) | MVBench | TempCompass | VideoMME (wo sub) | Avg | Avg (wo mra) |
|---|---|---|---|---|---|---|---|---|
| Qwen2.5-VL-7B | 28.1 (33.8/22.3) | 45.8 | 61.3 | 60.7 | 72.4 | 53.7 | 50.0 | 54.6 |
| Qwen2.5-VL-7B + 50% Pruning | **28.2 (34.5/21.9) [+0.1]** | 45.1 [–0.7] | 61.3 [+0.0] | 60.0 [–0.7] | **72.8 [+0.4]** | 52.8 [–0.9] | 49.8 [–0.2] | 54.4 [–0.2] |
| V-Reason-7B (Lite); (lr: 3e-4) | **30.5 (37.3/23.7) [+2.4]** | **46.7 [+1.6]** | **64.8 [+3.7]** | 60.6 [–0.1] | 72.3 [–0.1] | **53.5 [–0.2]** | **51.3 [+1.3]** | **55.9 [+1.3]** |

as compared to the **+16.9%** improvement obtained with Video-R1. Second, the Lite variant, while improving memory efficiency, incurs a measurable drop in accuracy for medium and long-duration videos, suggesting that pruning may discard valuable temporal information for those cases. Such limitations can be investigated as future work, as described next.

## G    FUTURE WORK

To our knowledge, V-Reason is the first work that targets the *video reasoning without training* problem. Hence, a number of exciting avenues of future research exist.

First, our entropy-based objective is applied only at inference time; integrating it into model training could potentially yield stronger gains and is an avenue for potential future research. Other directions of future research include exploring alternative inference-time metrics and loss functions that can further enhance reasoning.

Second, as a training-free framework, our method does not leverage task-specific supervision, which may limit its ability to capture nuanced reasoning strategies compared to reinforcement learning-based approaches. Hence, a combination of supervised finetuning and inference-time optimization-based reasoning techniques can also be explored in the future. Additionally, tailored solutions that can handle longer videos for the Lite variant can also be investigated.

Finally, although our proposed approach is motivated for videos, the idea of entropy-based inference-time optimization for enhanced reasoning is generic and can be extended to large language models (LLMs). We conducted a preliminary analysis of the entropy behavior of language models for MATH reasoning tasks and observed similar trends as the video models. We discuss the details below.

**LLM Entropy Curves.** Figure 6 shows the entropy curves of Qwen2.5 based LLMs averaged over a subset of 100 samples on the MATH dataset. It shows that better models have delayed peak and lower entropy overall. These trends are similar to those observed in the video modality suggesting that the proposed approach can also be extended to LLMs. We leave this for future work since it requires non-trivial contributions as the current approach cannot be directly applied due to the absence of the tokens to tune (akin to video tokens) in the inputs to LLMs. However, it is very exciting to see similar macro-exploration and macro-exploitation trends with cycles of micro-exploration and micro-exploitation that cause a delayed entropy maximum for the better model even for LLMs.

## H    ADDITIONAL QUALITATIVE RESULTS

Figures 7, 8, 9, and 10 shows additional examples where the baseline Qwen-2.5-VL-7B failed to arrive at the correct solution while V-Reason arrived at the correct answer following an alternative reasoning path.

Figures 11, 12, 13, and 14 show examples where both the baseline Qwen-2.5-VL-7B and V-Reason arrive at the correct answer while going through similar or alternative reasoning traces. In all these examples, V-Reason shows the consistent trend of longer exploration (delayed peak) and lower overall entropy induced by the micro-exploration and micro-exploitation cycles in our proposed optimization objective. In particular, Figure 11 shows that V-Reason arrives at the correct solution

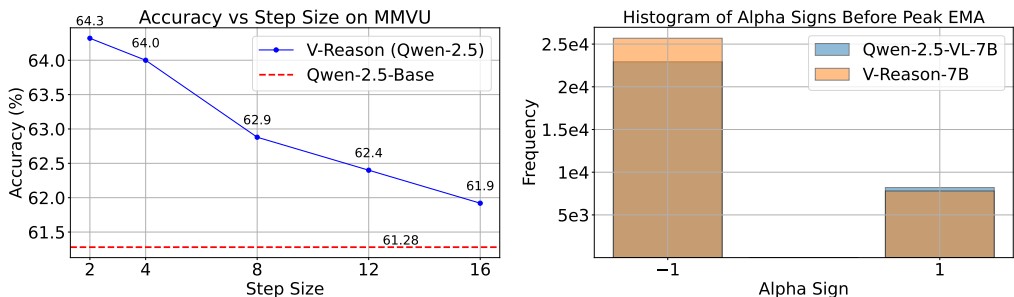

Figure 4: Optimization step-size ablations.    Figure 5: Alpha histogram before peak EMA entropy.

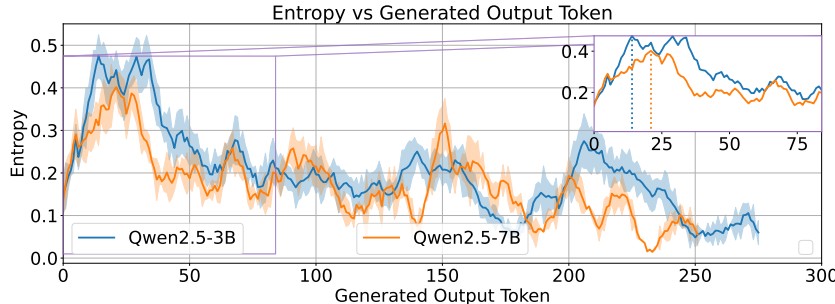

Figure 6: *LLM analysis*: (a) Entropy of the output distribution averaged over 100 samples of the MATH dataset (Yang et al., 2024a). Similar to Video LMMs, we see clear macro-exploration and macro-exploitation phases (having micro-exploration and micro-exploitations) with bigger, more accurate models showing lower overall entropy (lower and later peak, followed by a lower final entropy during the macro-exploitation). This shows that these key insights can be adopted for enhancing reasoning in LLMs too in a training-free way using an inference-time optimization technique.

using fewer output tokens as compared to the baseline as illustrated in Figure 1(d). Interestingly, in Figure 13 `V-Reason` uses more output tokens to provide the correct answer as compared to the baseline. However, `V-Reason` results in a more confident answer as seen from the lower overall entropy as compared to the baseline. This trend of higher confidence and lower overall entropy is seen in all the examples of `V-Reason` suggesting the effectiveness of our macro-exploitation phase induced by our proposed objective function.

# I    SUPPLEMENTAL INFORMATION: LLM USAGE

LLM was used to polish the writing (e.g., grammar, clarity). It was not used for literature retrieval, discovery of related work, or research ideation.

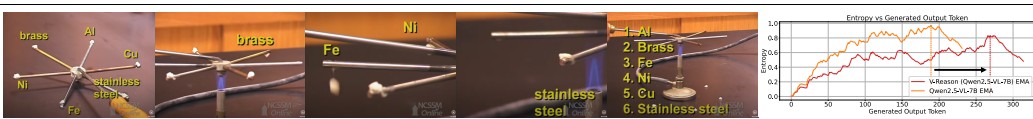

**Question:** Which of the following is the most accurate explanation for why the indicator on the aluminum rod melted first? A. Aluminum has a higher thermal conductivity than copper, allowing heat to travel faster along the rod. B. Aluminum has a lower heat capacity per unit volume than copper, so it heats up faster despite having a lower thermal conductivity. C. The melting point of aluminum is lower than that of copper, causing the indicator to melt sooner. D. Aluminum rods have a larger cross-sectional area, reducing thermal resistance. E. The experiment demonstrates experimental error; copper should have heated up faster in theory.

**Qwen-2.5-VL-7B:**

Let me think through this step by step. The question is asking why the indicator on the aluminum rod melted first. \n\nFirst, let's consider the options provided:\n- A. This suggests that aluminum's higher thermal conductivity allows heat to travel faster, which doesn't directly explain why the indicator melted first.\n- B. This option talks about heat capacity and thermal conductivity, which are important but doesn't directly address why the indicator melted first.\n- C. This option mentions the melting point of aluminum being lower than copper, which is a key factor. If aluminum melts at a lower temperature, it would indeed melt first.\n- D. This option talks about cross-sectional area, which is not directly related to the melting of the indicator.\n- E. This suggests that the experiment might have errors, but the question is asking for the most accurate explanation, not the validity of the experiment.\n\nGiven the context of the experiment, the most accurate explanation is related to the melting point of the materials. Since aluminum has a lower melting point than copper, the indicator on the aluminum rod would melt first.\n\nTherefore, the correct answer is:\n\n<answer>C</answer>

**V-Reason-7B:**

Let me think through this step by step. The question is asking why the indicator on the aluminum rod melted first. The key points to consider are the thermal conductivity, heat capacity, and melting point of aluminum and copper.\n\n1. **Thermal Conductivity**: Aluminum has a higher thermal conductivity than copper. This means heat travels faster through aluminum, but this doesn't directly explain why the indicator melted first. It's more about how quickly the heat is distributed along the rod.\n\n2. **Heat Capacity**: Heat capacity per unit volume is a factor here. Aluminum has a lower heat capacity per unit volume compared to copper. This means that for a given amount of heat, aluminum will heat up faster than copper. This is because it takes less energy to raise the temperature of aluminum compared to copper.\n\n3. **Melting Point**: The melting point of aluminum is lower than that of copper. However, this is not the primary reason for the indicator melting first. The melting point is a property of the material and doesn't directly relate to the rate at which the indicator melts.\n\n4. **Cross-sectional Area**: The cross-sectional area of the rods is not mentioned, so we can't assume it's a factor here.\n\n5. **Experimental Error**: The experiment is designed to measure the specific heat of metals, and the observed behavior is consistent with the properties of the materials.\n\nGiven these points, the most accurate explanation is that aluminum has a lower heat capacity per unit volume than copper, so it heats up faster despite having a lower thermal conductivity.\n\n<answer>B</answer>

Figure 7: Qualitative result: An example output and comparison with the baseline Qwen-2.5-VL-7B. Black arrow denotes the shift in the peak demonstrating longer exploration for V-Reason.

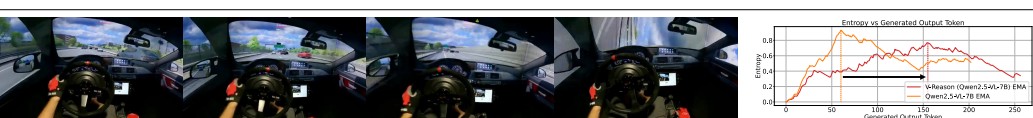

**Question:** What mechanical components are needed to achieve the actions in the video? A. recirculating-ball B. rack-and-pinion C. worm-and-roller D. planetary gear set E. A, B, and C

**Qwen-2.5-VL-7B:**

Let me think about this. The video shows a person driving a car, and the steering wheel is being turned. The options provided are related to different types of steering systems. \n\n- A recirculating-ball system is a type of steering gear that uses a ball nut and a ball screw to convert the rotary motion of the steering wheel into the linear motion of the steering shaft.\n- A rack-and-pinion system uses a rack and a pinion gear to convert the rotary motion of the steering wheel into the linear motion of the steering shaft.\n- A worm-and-roller system is another type of steering gear that uses a worm gear and a roller to convert the rotary motion of the steering wheel into the linear motion of the steering shaft.\n\nGiven that the video shows a person driving a car and the steering wheel is being turned, it is likely that the car is equipped with a rack-and-pinion steering system. This is the most common type of steering system used in modern cars.\n\n<answer>B</answer>

**V-Reason-7B:**

Let me think about this. The video shows a person driving a car, and the steering wheel is being turned. The steering system in a car can be either a recirculating-ball, a rack-and-pinion, or a worm-and-roller. The recirculating-ball and rack-and-pinion systems are more common in modern cars, while the worm-and-roller system is less common. The recirculating-ball system is known for its smooth operation and is often used in high-performance cars. The rack-and-pinion system is more common in everyday cars and is known for its efficiency and ease of use. The worm-and-roller system is less common and is more complex.\n\nGiven the context of the video, it seems to be a high-performance car, which would likely use a recirculating-ball system. However, without more specific information, it's difficult to be certain. Let's break it down: the recirculating-ball system is known for its smooth operation and is often used in high-performance cars, which matches the context of the video. The rack-and-pinion system is more common in everyday cars and is known for its efficiency and ease of use. The worm-and-roller system is less common and is more complex.\n\n<answer>E</answer>

Figure 8: Qualitative result: An example output and comparison with the baseline Qwen-2.5-VL-7B. Black arrow denotes the shift in the peak demonstrating longer exploration for V-Reason.

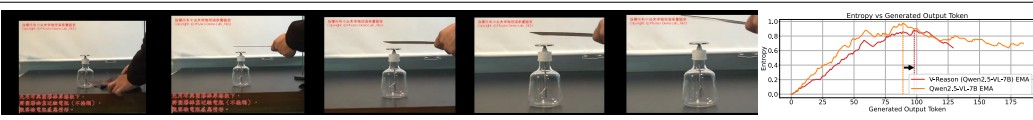

**Question:** What is the explanation of the shown phenomenon? A. Because the plastic rod is charged, after direct contact with the iron pieces, the iron pieces acquire opposite charges, causing them to attract each other. B. Because the plastic rod is charged and is not in direct contact with the iron pieces, the iron pieces are inductively charged and carry the same charge and repel each other. C. Because the air ambient temperature is high, the iron sheets expand and separate when heated. D. Because the plastic rod has a high temperature, the heat is transferred to the iron sheets causing them to expand and separate. E. Because the plastic rod is magnetized, and when brought near the iron pieces, induces magnetic forces causing the iron pieces to attract each other.

**Qwen-2.5-VL-7B:**

Let me think about this. The video shows a plastic rod being rubbed with a cloth and then brought near a set of iron pieces inside a bell jar. The iron pieces are observed to move, indicating a force of attraction or repulsion. The text in the video suggests that the rod is charged and not in direct contact with the iron pieces, which aligns with the concept of electrostatic induction.\n\nThe phenomenon described in the video is consistent with the principle of electrostatic induction. When a charged object (in this case, the plastic rod) is brought near uncharged objects (the iron pieces), it induces a charge on the iron pieces. If the rod is positively charged, it will induce a negative charge on the iron pieces that are closer to it, and a positive charge on the iron pieces that are farther away. This causes the iron pieces to repel each other.\n\nGiven this reasoning, the correct explanation is:\n\nB

**V-Reason-7B:**

Let me think about this. The video shows a plastic rod being brought near iron pieces without touching them, and the iron pieces move apart. This phenomenon is not due to temperature or magnetism, as the rod is not hot and the iron pieces do not show magnetic attraction. The rod is not in direct contact with the iron pieces, so it cannot transfer heat or magnetism. The iron pieces are not expanding due to heat, as they are not in a high-temperature environment. The only logical explanation is that the rod is charged and induces a charge in the iron pieces, causing them to repel each other.\n\n<answer>B</answer>

Figure 9: Qualitative result: An example output and comparison with the baseline Qwen-2.5-VL-7B. Black arrow denotes the shift in the peak demonstrating longer exploration for `V-Reason`.

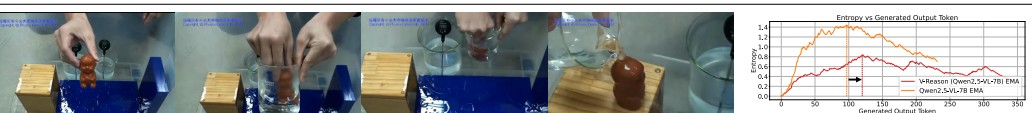

**Question:** Which of the following best explains the phenomenon on the beaker in the video? A. The internal air pressure increases to about 1.05 atm due to thermal expansion; this pressure increase forces warm air out of the object, cooling the surrounding air and causing condensation on the beaker. B. The internal air pressure decreases to about 0.90 atm due to thermal contraction; this pressure drop causes air to flow into the object, warming the surrounding air and preventing condensation on the beaker. C. The internal air pressure remains at 1 atm because the temperature change is insufficient to affect the pressure significantly; condensation occurs due to humidity changes in the room. D. The internal air pressure decreases to about 0.94 atm due to thermal contraction; the pressure difference draws moist ambient air toward the cold surface of the beaker, where it cools below the dew point and condenses. E. The internal air pressure decreases slightly, but the condensation is primarily due to evaporation of the cold water cooling the beaker surface below the ambient dew point.

**Qwen-2.5-VL-7B:**

Let me think through this step by step. The video shows a beaker with a thermometer inside, and a person is manipulating a red object, possibly a hot object, near the beaker. The beaker is filled with water, and the thermometer is submerged in it. The red object is being moved in and out of the water, and the water level in the beaker changes slightly.\n\nThe phenomenon observed is likely related to the temperature change of the water and the resulting pressure changes. When the red object is moved into the water, it likely causes a temperature change in the water, which in turn affects the pressure inside the beaker.\n\nLet's break it down:\n- The red object is likely heated, causing the water to warm up.\n- As the water warms up, it expands, increasing the internal air pressure in the beaker.\n- This increased pressure forces the warm air out of the beaker, cooling the surrounding air and causing condensation on the beaker.\n\nGiven this reasoning, the best explanation is that the internal air pressure increases due to thermal expansion, forcing warm air out and causing condensation.\n\n<answer>A</answer>

**V-Reason-7B:**

Let me think through this step by step. The video shows a beaker with a condensation phenomenon occurring on its surface. Condensation typically happens when the temperature of the beaker is lower than the surrounding air, causing the moisture in the air to condense on the beaker's surface. This suggests a temperature difference between the beaker and the surrounding air.\n\nNow, let's consider the options:\n\nA. This option talks about thermal expansion and condensation due to air pressure changes. However, the video doesn't show any significant change in the beaker's temperature or pressure that would cause such a phenomenon.\n\nB. This option suggests a decrease in internal air pressure due to thermal contraction, which would cause air to flow into the object, warming the surrounding air and preventing condensation. This doesn't align with the observed condensation.\n\nC. This option states that the internal air pressure remains at 1 atm, which is unlikely given the temperature change. Also, it doesn't explain the condensation.\n\nD. This option explains that the internal air pressure decreases due to thermal contraction, drawing moist ambient air toward the cold surface of the beaker, where it cools below the dew point and condenses. This aligns well with the observed condensation.\n\nE. This option suggests that the condensation is primarily due to evaporation of the cold water cooling the beaker surface, which doesn't explain the condensation on the beaker's surface.\n\nAfter considering all the options, the most logical explanation is that the internal air pressure decreases due to thermal contraction, drawing moist ambient air toward the cold surface of the beaker, where it cools below the dew point and condenses.\n\n<answer>D</answer>

Figure 10: Qualitative result: An example output and comparison with the baseline Qwen-2.5-VL-7B. Black arrow denotes the shift in the peak demonstrating longer exploration for `V-Reason`.

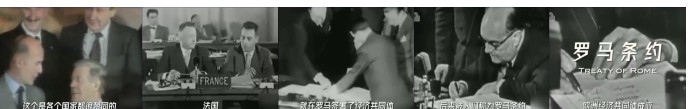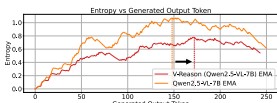

**Question**: Which of the following best explains the primary benefits brought by the treaty shown in the video? A. By creating a common market allows free movement of goods, services, capital, and labor, letting members specialize by their strengths. B. By adopting a single currency to eliminate exchange rate fluctuations and reduce transaction costs among member states. C. By imposing uniform protectionist tariffs against non-member countries to shield domestic industries from external competition. D. By integrating fiscal policies to redistribute wealth from richer to poorer member states, reducing economic disparities. E. By standardizing labor laws and working conditions across member states to prevent a 'race to the bottom' in labor standards.

**Qwen-2.5-VL-7B**:

Let me think about this. The video seems to be discussing the Treaty of Rome, which is the founding treaty of the European Economic Community (EEC), the predecessor of the European Union (EU). The treaty aimed to create a common market and economic union among its member states. \n\nOption A talks about the creation of a common market allowing free movement of goods, services, capital, and labor, which aligns with the goal of the EEC to integrate economies and facilitate trade among member states.\n\nOption B mentions adopting a single currency, which was not part of the initial EEC treaty but became a part of the later European Union.\n\nOption C is about protectionist tariffs, which is not a primary benefit of the EEC treaty. The EEC aimed for free trade among member states. \n\nOption D is about fiscal policies and wealth redistribution, which is not a primary focus of the EEC treaty.\n\nOption E is about standardizing labor laws, which was also not a primary focus of the EEC treaty.\n\nGiven the context of the video and the options provided, the primary benefit brought by the treaty shown in the video is the creation of a common market allowing free movement of goods, services, capital, and labor.\n\n\<answer>A</answer>

**V-Reason-7B**:

Let me think about this. The video is about the Treaty of Rome, which is the founding treaty of the European Economic Community (EEC), the precursor to the European Union (EU). The primary benefits of such a treaty would be related to the creation of a common market and the facilitation of trade and economic integration among member states.\n\nOption A talks about the creation of a common market that allows for the free movement of goods, services, capital, and labor. This aligns well with the goals of the EEC, which aimed to create a single market and a customs union.\n\nOption B mentions adopting a single currency. While the EU later adopted the euro, this was not part of the original Treaty of Rome.\n\nOption C is about protectionist tariffs, which is the opposite of what the EEC aimed for, as it sought to create a free trade area.\n\nOption D is about fiscal policies and wealth redistribution, which is not a primary benefit of the EEC.\n\nOption E is about standardizing labor laws, which is also not a primary benefit of the EEC.\n\nGiven this analysis, the best explanation for the primary benefits brought by the treaty shown in the video is:\n\n\<answer>A</answer>

Figure 11: Qualitative result: An example output and comparison with the baseline Qwen-2.5-VL-7B. Black arrow denotes the shift in the peak demonstrating longer exploration for `V-Reason`.

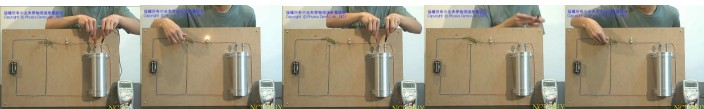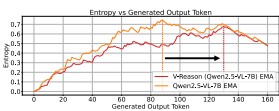

**Question**: Which of the following best explains the effect of increasing resistance in the circuit on the voltmeter readings over time? A. The increased resistance causes the capacitor to charge faster, so the voltmeter reading increases more rapidly. B. The increased resistance causes the capacitor to charge more slowly initially, but the final voltage reached is higher. C. The increased resistance has no effect on the charging process or the voltmeter readings since the battery voltage is constant. D. The increased resistance causes the capacitor to charge more slowly, so the voltmeter reading increases more slowly. E. The increased resistance causes the capacitor to discharge, causing the voltmeter reading to decrease over time.

**Qwen-2.5-VL-7B**:

Let me think about this. When resistance increases in a circuit, it affects the rate at which the capacitor charges. The voltmeter measures the voltage across the capacitor. If the resistance increases, the current through the circuit decreases, which means the capacitor charges more slowly. This slower charging process would result in a slower increase in the voltage reading on the voltmeter.\n\nNow, let's break it down:\n- Increasing resistance means less current flows through the circuit.\n- Less current means the capacitor charges more slowly.\n- Slower charging means the voltage on the voltmeter increases more slowly.\n\nGiven these points, the correct answer should reflect that the increased resistance causes the capacitor to charge more slowly, leading to a slower increase in the voltmeter reading.\n\n\<answer>D</answer>

**V-Reason-7B**:

Let me think about this. Increasing resistance in a circuit generally means that the current flowing through the circuit will decrease. This is because resistance is a measure of how much a material opposes the flow of electric current. When the resistance increases, the current decreases, which in turn affects the rate at which the capacitor charges.\n\nNow, let's consider the effect on the voltmeter readings. The voltmeter measures the voltage across the capacitor. When the resistance increases, the current decreases, and this means that the capacitor will charge more slowly. As a result, the voltage across the capacitor will increase more slowly as well.\n\nSo, the correct option should reflect that the increased resistance causes the capacitor to charge more slowly, leading to a slower increase in the voltmeter reading.\n\n\<answer>D</answer>

Figure 12: Qualitative result: An example output and comparison with the baseline Qwen-2.5-VL-7B. Black arrow denotes the shift in the peak demonstrating longer exploration for `V-Reason`.

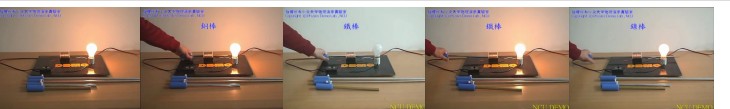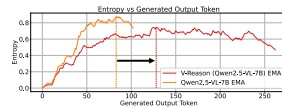

**Question**: Which of the following best explains the effect of increasing resistance in the circuit on the voltmeter readings over time? A. The increased resistance causes the capacitor to charge faster, so the voltmeter reading increases more rapidly. B. The increased resistance causes the capacitor to charge more slowly initially, but the final voltage reached is higher. C. The increased resistance has no effect on the charging process or the voltmeter readings since the battery voltage is constant. D. The increased resistance causes the capacitor to charge more slowly, so the voltmeter reading increases more slowly. E. The increased resistance causes the capacitor to discharge, causing the voltmeter reading to decrease over time.

**Qwen-2.5-VL-7B:**

Let me think about this. When resistance increases in a circuit, it affects the rate at which the capacitor charges. The voltmeter measures the voltage across the capacitor. If the resistance increases, the current through the circuit decreases, which means the capacitor charges more slowly. This slower charging process would result in a slower increase in the voltage reading on the voltmeter.\n\nNow, let's break it down:\n- Increasing resistance means less current flows through the circuit.\n- Less current means the capacitor charges more slowly.\n- Slower charging means the voltage on the voltmeter increases more slowly.\n\nGiven these points, the correct answer should reflect that the increased resistance causes the capacitor to charge more slowly, leading to a slower increase in the voltmeter reading.\n\n<answer>D</answer>

**V-Reason-7B:**

Let me think about this. Increasing resistance in a circuit generally means that the current flowing through the circuit will decrease. This is because resistance is a measure of how much a material opposes the flow of electric current. When the resistance increases, the current decreases, which in turn affects the rate at which the capacitor charges.\n\nNow, let's consider the effect on the voltmeter readings. The voltmeter measures the voltage across the capacitor. When the resistance increases, the current decreases, and this means that the capacitor will charge more slowly. As a result, the voltage across the capacitor will increase more slowly as well.\n\nSo, the correct option should reflect that the increased resistance causes the capacitor to charge more slowly, leading to a slower increase in the voltmeter reading.\n\n<answer>D</answer>

Figure 13: Qualitative result: An example output and comparison with the baseline Qwen-2.5-VL-7B. Black arrow denotes the shift in the peak demonstrating longer exploration for V-Reason.

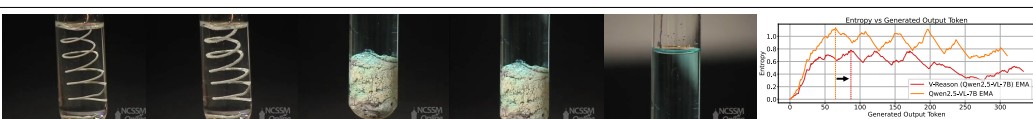

**Question**: If the solution in the video is a salt solution, what is the most likely chemical reaction equation that would occur in the test tube? A. $Cu + Pb(NO_3)_2 \rightarrow Cu(NO_3)_2 + Pb$ B. $Cu + 2AgNO_3 \rightarrow Cu(NO_3)_2 + 2Ag$ C. $Cu + FeSO_4 \rightarrow CuSO_4 + Fe$ D. $Cu(NO_3)_2 + Na_2S \rightarrow CuS + 2NaNO_3$ E. $CuCl_2 + Na_2SO_4 \rightarrow CuSO_4 + 2NaCl$

**Qwen-2.5-VL-7B:**

Let me think about this. The video shows a test tube with a wire that appears to be copper, and the solution is a salt solution. The wire seems to be reacting with the solution, and there's a visible change in the wire's surface, indicating a chemical reaction.\n\nThe options provided are all potential reactions involving copper. Let's break it down:\n\nA. This reaction involves copper reacting with lead nitrate to form copper nitrate and lead. This doesn't seem to fit the visual evidence since the wire doesn't appear to be lead.\n\nB. This reaction involves copper reacting with silver nitrate to form copper nitrate and silver. This could be a plausible reaction, as silver nitrate is a common reagent in chemistry labs and could cause a visible change in the wire's surface.\n\nC. This reaction involves copper reacting with iron sulfate to form copper sulfate and iron. This is less likely because iron is typically more reactive than copper and would likely be the one to react first.\n\nD. This reaction involves copper nitrate reacting with sodium sulfide to form copper sulfide and sodium nitrate. This is also less likely because copper nitrate is not typically used in this reaction.\n\nE. This reaction involves copper chloride reacting with sodium sulfate to form copper sulfate and sodium chloride. This is less likely because copper chloride is not typically used in this reaction.\n\nGiven the visual evidence and the options provided, the most likely reaction is the one that involves copper reacting with a silver nitrate solution, which is option B.\n\n<answer>B</answer>

**V-Reason-7B:**

Let me think about this. The video shows a test tube with a wire that appears to be copper, and the solution is a salt solution. The reaction seems to involve the copper wire dissolving and forming a precipitate at the bottom of the tube.\n\n\n\nOption A involves copper reacting with lead nitrate to form copper nitrate and lead. This doesn't seem to fit the observation in the video as there's no lead nitrate present.\n\nOption B involves copper reacting with silver nitrate to form copper nitrate and silver. This could be a possibility as silver nitrate is a common salt solution and could react with copper.\n\nOption C involves copper reacting with iron(II) sulfate to form copper(II) sulfate and iron. This is less likely as iron(II) sulfate is not a common salt solution and the reaction would not form a precipitate at the bottom.\n\nOption D involves copper(II) nitrate reacting with sodium sulfide to form copper sulfide and sodium nitrate. This could be a possibility as copper(II) nitrate is a common salt solution and could react with sodium sulfide to form a precipitate.\n\nOption E involves copper(II) chloride reacting with sodium sulfate to form copper(II) sulfate and sodium chloride. This is less likely as sodium sulfate is not a common salt solution and the reaction would not form a precipitate at the bottom.\n\nGiven the observation of a precipitate forming at the bottom of the tube, option B seems the most likely as it involves a common salt solution (silver nitrate) and could form a precipitate (silver) at the bottom of the tube.\n\n<answer>B</answer>

Figure 14: Qualitative result: An example output and comparison with the baseline Qwen-2.5-VL-7B. Black arrow denotes the shift in the peak demonstrating longer exploration for V-Reason.

