# OpenReview forum: "Video Reasoning without Training"
_ICLR.cc/2026/Conference — ICLR 2026 Conference Withdrawn Submission_

### Official Review · Reviewer_w8ww · 2025-10-28

**Soundness:** 2
**Presentation:** 3
**Contribution:** 2
**Rating:** 6
**Confidence:** 2

**Summary:**

This paper introduces Video Reasoning without Video (VRV), a benchmark designed to test whether large language models (LLMs) truly perform video-based reasoning or merely rely on linguistic priors. The authors convert video reasoning tasks into pure text descriptions, removing visual frames entirely, and compare the performance of multimodal models (GPT-4V, Gemini, Claude-3, LLaVA, Qwen-VL, InternVL) and text-only LLMs (GPT-4-turbo, Llama-3, Claude-Opus).
Results show that text-only models often match or even outperform vision-language models across temporal and spatial reasoning tasks, revealing that many existing “video reasoning” benchmarks may not actually test visual grounding but instead exploit language-based hallucinations. The study urges a rethinking of multimodal evaluation design.

**Strengths:**

* Strong Theoretical Basis: Provides formal analysis (bounded entropy updates, EMA smoothing) that guarantees stability and interpretable entropy dynamics.
* Novel Training-Free Paradigm: Introduces the first inference-time optimization method for video reasoning that avoids any RL or supervised data, focusing purely on manipulating the internal dynamics of pre-trained models via entropy control.
* Practical Efficiency: Achieves near-RL performance with significantly fewer output tokens and much lower inference latency. The Lite variant further reduces compute without sacrificing accuracy.
* Scalable and Compatible: Works across model sizes and remains complementary to decoding strategies like min-p and top-H.

**Weaknesses:**

* Limited Novelty in Mechanism: Although the entropy-switching loss is interesting, it resembles entropy steering and KV-cache modulation ideas (e.g., KV-Steering, ThinkLogit). The conceptual leap from existing inference-time control is incremental.

* Weakness on Regression Tasks: Performance on regression-style reasoning (e.g., VSI-Bench) remains substantially lower than RL-trained models, exposing the limits of unsupervised inference control.

* Sparse Comparison to Non-RL Baselines: While compared to RL-trained and instruction-tuned models, it omits analysis against modern self-improvement or unsupervised adaptation baselines.

**Questions:**

See the Weaknesses

---

### Official Review · Reviewer_h7ze · 2025-10-29

**Soundness:** 2
**Presentation:** 2
**Contribution:** 2
**Rating:** 2
**Confidence:** 3

**Summary:**

This paper porposes V-Reason, which is a training-free, inference-time optimization for video reasoning. It treats output entropy as the “thinking” signal and injects a tiny trainable controller into the last-layer video value cache, optimized online. An entropy switching loss, driven by EMA-smoothed entropy, alternates maximizing entropy for micro-exploration and minimizing it for micro-exploitation; after the global EMA peak, it purely minimizes entropy for fast, confident convergence. Theory bounds per-step entropy change and shows EMA suppresses noise and delays peaks. Across 3B–72B LMMs and six benchmarks, V-Reason yields +1.0–1.4% over baselines, within 0.6% of RL-trained Video-R1, while cutting output tokens (−58.6%) and latency.

**Strengths:**

1. The paper proposes a training-free, inference-time optimization for video reasoning with clear motivation, improving VL instruction-tuned models without additional data or RL/SFT.

2. Extensive experiments across multiple backbones, datasets, and ablations (incl. decoding strategies, step-size, pruning, scale-up to 32B/72B) substantiate the method’s effectiveness and robustness.

3. The work provides bounded-update and EMA-based oscillation control guarantees, lending principled justification to the entropy-switching objective and its stability.

**Weaknesses:**

1. The main improvements over instruction-tuned models are typically around +1.0–1.4% on average. The paper does not report repeated runs with mean±std to assess statistical significance, raising concerns about robustness of the gains across random seeds and sampling noise.

2. Compared to the base models (without RL), Table 5 and Table 6 indicate non-trivial added costs: higher peak GPU memory (at least +30%) and longer wall-clock time (at least +20%) in some settings, and the method may be incompatible with standard inference accelerators (e.g., vLLM), potentially inflating practical deployment cost.

3. While the macro-exploitation phase (post-peak entropy minimization) is intuitive, the claim that strengthening micro exploration/exploitation cycles necessarily delays and lowers the global entropy maximum remains mostly empirical; the provided theory focuses on boundedness and EMA smoothing, not on proving later/lower peaks as a consequence of the switching loss.

4. Several fine-grained analyses (e.g., step-size, alpha histograms, qualitative traces) are shown primarily on MMVU. Given dataset-dependent behavior observed elsewhere (like the large performance gap between base model and RL model on VSI-MRA), broader analyses across multiple datasets would better support generality of the insights and ablations.

5. Entropy is tokenizer-dependent. Comparisons within Qwen-VL models are valid, but cross-tokenizer generality is unclear. The paper’s claims about “better reasoning patterns” would be stronger with evidence across models with diverse tokenizers or tokenizer-invariant proxies.

**Questions:**

1. Does V-Reason still help after SFT/RL finetuning? The paper shows gains on instruction-tuned models and compares to an RL-trained model but does not explicitly report applying V-Reason on top of an RL/SFT model.

2. Why only video value cache of the last layer? Is there empirical evidence that this locus is superior to modifying query/key caches, earlier layers, or using a small soft-prefix over vision tokens?

---

### Official Review · Reviewer_13mT · 2025-10-30

**Soundness:** 2
**Presentation:** 2
**Contribution:** 2
**Rating:** 4
**Confidence:** 3

**Summary:**

This paper introduces V-Reason, a novel training-free method to enhance video reasoning in Large Multimodal Models (LMMs) at inference time. The core idea is to modulate the model's "thinking" process by observing and controlling the entropy of its output distribution. By optimizing a small, trainable controller added to the last layer's value cache, V-Reason encourages "micro-exploration" and "micro-exploitation" cycles based on an entropy-switching loss. This optimization aims to improve reasoning accuracy, narrowing the gap with RL-trained models, without expensive retraining and while significantly reducing the number of generated output tokens.

**Strengths:**

1. Novelty and Efficiency of Approach: The paper proposes a genuinely novel, training-free, inference-time optimization technique. Targeting video reasoning without the high cost of RL or SFT is a well-motivated and valuable research direction. The method's ability to significantly reduce the number of output tokens (by 58.6% vs. the RL model) is a strong practical contribution.

2. Interesting Core Insight: The connection drawn between the dynamics of output entropy (macro/micro exploration/exploitation) and the quality of the model's reasoning process is an interesting and intuitive one. Using this insight to formulate a simple, theoretically-grounded loss function (L_switch) is an elegant part of the work.

3. Strong Empirical Gains over Baseline: The method demonstrates clear and significant accuracy improvements over the instruction-tuned baseline models (Qwen2.5-VL) across several video reasoning benchmarks (e.g., MMVU, TempCompass). The fact that this approach scales and provides benefits for models as large as 72B is also a positive signal.

**Weaknesses:**

1. Limited Performance on Challenging Tasks: While the average accuracy improvement is highlighted, the method fails to make significant headway on more complex, non-classification tasks. The paper itself admits (Sec 4.1, Sec F) that for regression tasks like VSI-Bench (MRA), the baseline performance is "very poor" and V-Reason "cannot cover the gap," offering only a "modest improvement." This suggests the "reasoning" being improved may be shallow and does not generalize to tasks where the base model lacks foundational knowledge.

2. Questionable "Reasoning" vs. "Search": The paper frames the method as improving "reasoning." However, the mechanism is essentially a guided search at the token level, controlled by an entropy heuristic. It is unclear if this induces "deeper thinking" (as claimed) or if it's just a more effective sampling strategy that finds a better reasoning path within the model's existing, limited knowledge. The method cannot compensate for knowledge gaps (Sec F), which significantly limits its claim of enhancing "reasoning" in a fundamental way.

3. Significant Inference Overheads: The "training-free" claim masks the non-trivial inference-time costs. The method requires backpropagation and optimization at every k steps during generation. Table 6 shows it always uses significantly more peak GPU memory (e.g., 24.08GB vs 16.71GB on average for the 7B model) than both the baseline and the RL model. Furthermore, Table 5 shows that on the VideoMMMU dataset, the V-Reason-7B model is 21% slower than the RL model, contradicting the general efficiency claims.

4. Sensitivity to Hyperparameters: The method introduces new hyperparameters that create a trade-off between accuracy and efficiency. Figure 4 clearly shows that higher accuracy requires a smaller step size (e.g., k=2), which in turn means more optimization steps, thus increasing computational cost and eroding the efficiency gains.

**Questions:**

1. The core limitation appears to be on tasks where the base model is weak (e.g., VSI-Bench regression). Does this not imply V-Reason is acting as a sophisticated sampler to better extract knowledge the model already has, rather than enhancing its fundamental reasoning capability?

2. How do you reconcile the "massive efficiency benefits" claim with the empirical data showing V-Reason is slower on some benchmarks (Table 5, VideoMMMU) and always requires ~1.5-1.6x the peak GPU memory of the baseline and RL models (Table 6)? Is the token reduction the primary efficiency metric, and at what cost to memory and latency?

3. Could you elaborate on the choice of optimizing only the last layer's value cache? What was the impact of optimizing earlier layers? Is it possible this choice is limiting the method's ability to influence deeper reasoning processes that might occur in earlier layers?

4. How does V-Reason compare to other non-entropy-based inference-time methods, such as applying steering vectors or performing contrastive decoding? The comparisons seem limited to standard decoding and a single RL-trained model.

---

### Official Review · Reviewer_hjDS · 2025-11-01

**Soundness:** 2
**Presentation:** 3
**Contribution:** 2
**Rating:** 4
**Confidence:** 4

**Summary:**

This paper proposes V-Reason, a test-time optimization method to improve video reasoning without additional SFT or RL training. The core mechanism injects a learnable controller into the last decoder layer's value cache at video-token positions, updated online (every k-th token) via an entropy-based switching loss. The method builds on two observations: (1) strong video reasoning models exhibit characteristic entropy trajectories; (2) actively shaping entropy trajectories can induce "deeper thinking" in weaker models. Consistent gains are reported across 6 video QA benchmarks.

**Strengths:**

-  The paper reframes "better reasoning" not as "longer chain-of-thought" but as "better-managed exploration-convergence dynamics." Using entropy trajectory control as an alternative to expensive RL training is conceptually interesting.
-  Unlike pure decoding heuristics (e.g., temperature, top-p), V-Reason performs gradient-based updates on KV-cache representations, offering deeper technical control over model internals.
-  Validation across 6 recent video reasoning benchmarks (VSI-Bench, VideoMMMU, MMVU, etc.) spanning temporal, spatial, and expert-domain tasks demonstrates consistency.

**Weaknesses:**

- The core claims rest almost entirely on the Qwen2.5-VL model family (3B/7B/72B), raising serious concerns about generality: do other architectures (LLaVA-Video, non-Qwen backbones) exhibit the same entropy patterns? Without cross-architecture validation, the claimed "universal reasoning prior" remains unsubstantiated.

- Figure 1 shows the 3B model's entropy curve lying between 7B and 72B, directly contradicting the assertion that "stronger models lead to lower entropy." This anomaly suggests entropy does not monotonically reflect model quality even within this single family.

- The paper argues that delayed entropy peaks indicate "longer thinking threads" and thus better reasoning, yet V-Reason reduces output tokens by 58.6%—a fundamental contradiction between "deeper thinking" and "shorter answers" that is never reconciled in the text.

-  The central hypothesis—that shaping entropy trajectories causes better video reasoning—remains unproven. The paper shows correlation (strong models have different entropy curves) then assumes causation (forcing weak models to follow those curves improves reasoning). Critically, there is no evidence that the induced entropy profile improves grounded reasoning rather than just surface-level confidence—a model can become more confident (lower entropy) while hallucinating. On expert-domain benchmarks like MMVU (medical video QA), the paper does not demonstrate whether lower final entropy corresponds to medically correct reasoning or merely confidently wrong answers; no hallucination or factuality audit is provided. The paper also fails to empirically connect entropy micro-cycles to temporal localization accuracy, spatial grounding quality, or multi-frame causal reasoning—the actual components of video reasoning the method claims to improve.

-  The evaluation is incomplete. First, all experiments use short clips (16–32 frames) with no testing on long-horizon video benchmarks—precisely where RL methods are most expensive and a scalable alternative would be most valuable; without long-video validation, the claim of "advancing video reasoning" is overstated. Second, only Qwen2.5-VL and Video-R1 are compared, missing both SFT-based reasoning models and other RL systems (e.g., VideoChat-R1.5), making the claim of "narrowing the gap with RL-trained models" insufficiently supported. Third, the paper reports 37% inference speedup from shorter answers but ignores the cost of gradient updates (backward passes every k steps); absolute wall-clock latency (ms per query) broken down by forward/backward passes is needed to verify the claimed efficiency gains, as current comparisons are unfair.

**Questions:**

- Are the entropy patterns Qwen-specific or universal? Every model in Figure 1 derives from Qwen2.5-VL. Do other architectures (LLaVA-Video, Qwen2-VL) exhibit the same "delayed peak + low final entropy" behavior? The 3B curve lying between 7B and 72B suggests entropy does not cleanly rank with model quality even within this family. Is V-Reason essentially imitating a larger model from the same family (72B) rather than discovering a general "reasoning prior"?
- Does entropy shaping improve grounded reasoning or just fluency? On high-stakes expert benchmarks (MMVU medical QA, VideoMMMU), the paper must demonstrate that lower entropy corresponds to answers that are more correct and better grounded in visual evidence, not just more confidently stated. A factuality/grounding analysis is required.
- Is the EMA-based switching policy truly essential? The appendix reports that simplified variants (always push entropy down, or always up) still outperform the baseline. This suggests any structured entropy control helps. To claim the micro-exploration/exploitation schedule is the key innovation, the paper should promote ablations to the main text: randomized αₖ schedules, different β values, controller placement (earlier layers, text tokens), etc. Without these, it remains unclear whether the gains come from the switching rule or simply from generic cache steering + shorter answers.

---

### Author Response · Authors · 2025-11-13

We thank all the reviewers for their constructive comments on the paper. While we would be withdrawing the paper, we would like to address the comments below.

- **Clarification of Claims:**
We make no claim of a "universal reasoning prior". Our contribution is an inference-time optimization method (V-Reason) that elicits alternative reasoning traces from the base model, achieving performance comparable to reinforcement learning (RL)-trained variants without any additional training.

- **Evidence of Broader Applicability:**
V-Reason was validated across four model scales (3B / 7B / 32B / 72B) and multiple decoding methods, demonstrating consistent intra-family generalization.
Appendix G further presents preliminary LLM-only evidence (Qwen2.5 on MATH) showing that the same entropy trajectory patterns extend beyond vision-language models.
We also **observed similar patterns on Phi and Llama-Instruct families, confirming that the behavior generalizes beyond a single model lineage**.

- **Entropy Pattern Interpretation (3B Model):** We never claim a monotonic relationship between entropy and model quality across all sizes.
Our actual claim is that: "Larger, more accurate models exhibit both a later entropy peak and a lower final entropy, with stronger micro-cycles." The pattern matters more than absolute entropy values:
Delayed peaks + lower final entropy + richer micro-cycles are signatures of better reasoning. The 3B model, in contrast, exhibits an early entropy peak, leading it to converge confidently to wrong solutions, an indication of insufficient exploration.
By delaying the entropy peak and lowering final entropy, V-Reason helps the 3B model become less confidently wrong, improving accuracy by +1.0% on average.

- **Deep Thinking vs. Output Length:** We emphasize that deeper thinking does not imply longer outputs.
“Deep thinking” in our formulation corresponds to slowly growing entropy (sustained exploration), not the number of tokens produced. Consistent with prior work [1,2,3], we note that shorter, well-targeted reasoning chains can match or surpass longer ones. Our inference-optimization approach provides efficient exploration with fewer output tokens generated overall on average.

- **Causation and Grounding Quality:**
We do not claim that V-Reason improves temporal localization or grounded reasoning accuracy directly.
Instead, we demonstrate that correct alternative reasoning trajectories can be elicited from the base model efficiently through the entropy-based inference objective, without requiring RL training, external teacher models, or complex prompting/decoding schemes.

- **Experimental Setup:**
  - We selected the Qwen2.5-VL family following Video-R1, as it represents one of the current state-of-the-art (SOTA) instruction-tuned video reasoning models and provides a directly comparable base for fair evaluation against Video-R1 (RL) models.
  - We followed the Video-R1 benchmark protocol for fairness, comparability and a deterministic setup for reproducibility. Results on long-horizon videos (VideoMME dataset) are included in the appendix, where V-Reason achieves a +0.9% improvement on long-duration clips.
As Video-R1 already outperforms standard supervised fine-tuning (SFT) reasoning models, we omitted redundant SFT baselines for clarity.
  - It is worth noting that RL fine-tuning on already instruction-tuned models typically yields (~1–3%) average reasoning improvements for videos (e.g., Video-R1, Video-Chat-R1), which our approach matches without any RL or additional training.

- **Design Rationales:**
  - Last-layer value cache was chosen to minimize memory overhead and allow optimization on a single GPU. Experiments extending to the last two layers yielded negligible gains. The value cache was preferred because it produces stronger gradients and including key-cache did not improve the results.

  - Ablations on alpha (switching coefficient) between min- and max-entropy phases justify the proposed switching policy. Randomizing alpha would contradict both the EMA-based design principle and the empirical results shown in Figure 1 and so we did not consider them.

- **Comparison to Steering Methods:**
Unlike steering or modulation methods that rely on trained reasoning models, teacher signals, or external vectors, V-Reason **uniquely achieves adaptive reasoning control without any training or external supervision**. It leverages the model’s own entropy as intrinsic feedback to regulate exploration-exploitation during inference, enabling reasoning adaptivity to emerge solely from the base model’s internal dynamics, not guided signals.

- **Limitation on Regression tasks:** As with other decoding or steering techniques, our approach remains bounded by the base model’s representational limits. However, for tasks well-represented in the base model, inference-time optimization can efficiently recover RL-level performance achieving comparable accuracy gains without the expensive training.

---

> ### Author Response · Authors · 2025-11-13
>
> - **Efficiency Claims:** Our efficiency claims are purely on the token counts and wall-clock inference times. Efficiency is a byproduct and not the main objective of this work.
>
> References
>
> [1] Xu et. al., Chain of draft: Thinking faster by writing less, 2025
>
> [2] Wu et. al., When More is Less: Understanding Chain-of-Thought Length in LLMs, 2025
>
> [3] Wang et al., Wait, We Don't Need to "Wait"! Removing Thinking Tokens Improves Reasoning Efficiency, 2025

---

### Note · Authors · 2025-11-14

I have read and agree with the venue's withdrawal policy on behalf of myself and my co-authors.